# *Staphylococcus aureus* FtsZ and PBP4 bind to the conformationally dynamic N-terminal domain of GpsB

**Michael D Sacco**[1†], **Lauren R Hammond**[2†], **Radwan E Noor**[3],
**Dipanwita Bhattacharya**[2], **Lily J McKnight**[2], **Jesper J Madsen**[1,4], **Xiujun Zhang**[1],
**Shane G Butler**[1], **M Trent Kemp**[1], **Aiden C Jaskolka-Brown**[1], **Sebastian J Khan**[2],
**Ioannis Gelis**[3], **Prahathees Eswara**[2*], **Yu Chen**[1*]

[1]Department of Molecular Medicine, Morsani College of Medicine, University of South Florida, Tampa, United States; [2]Department of Molecular Biosciences, University of South Florida, Tampa, United States; [3]Department of Chemistry, University of South Florida, Tampa, United States; [4]Global and Planetary Health, College of Public Health, University of South Florida, Tampa, United States

**\*For correspondence:**
eswara@usf.edu (PE);
ychen1@usf.edu (YC)

[†]These authors contributed equally to this work

**Competing interest:** The authors declare that no competing interests exist.

**Abstract** In the Firmicutes phylum, GpsB is a membrane associated protein that coordinates peptidoglycan synthesis with cell growth and division. Although GpsB has been studied in several bacteria, the structure, function, and interactome of *Staphylococcus aureus* GpsB is largely uncharacterized. To address this knowledge gap, we solved the crystal structure of the N-terminal domain of *S. aureus* GpsB, which adopts an atypical, asymmetric dimer, and demonstrates major conformational flexibility that can be mapped to a hinge region formed by a three-residue insertion exclusive to *Staphylococci*. When this three-residue insertion is excised, its thermal stability increases, and the mutant no longer produces a previously reported lethal phenotype when overexpressed in *Bacillus subtilis*. In *S. aureus*, we show that these hinge mutants are less functional and speculate that the conformational flexibility imparted by the hinge region may serve as a dynamic switch to fine-tune the function of the GpsB complex and/or to promote interaction with its various partners. Furthermore, we provide the first biochemical, biophysical, and crystallographic evidence that the N-terminal domain of GpsB binds not only PBP4, but also FtsZ, through a conserved recognition motif located on their C-termini, thus coupling peptidoglycan synthesis to cell division. Taken together, the unique structure of *S. aureus* GpsB and its direct interaction with FtsZ/PBP4 provide deeper insight into the central role of GpsB in *S. aureus* cell division.

## Editor's evaluation

This valuable work reports a unique N-terminal motif of *Staphylococcus aureus* GpsB, the co-crystal structure of GpsB with the C-terminus of PBP4, and the direct interaction between GpsB and the C-terminus of FtsZ. The evidence supporting these discoveries is convincing, with biochemical and structural characterizations. This study sheds light on the role of GpsB in the cell division of this important pathogen.

## Introduction

Bacterial cell division is a dynamic process involving more than a dozen proteins that form a multimeric complex at mid-cell. Collectively known as the divisome, this network of scaffolding proteins and enzymes stimulates peptidoglycan synthesis, constricts the existing membrane, and forms the septal

cell wall (*Mahone and Goley, 2020*). While divisomal proteins may differ among certain clades, most are conserved among all bacteria. Perhaps the most important and well-studied divisomal protein is FtsZ, a bacterial homolog of eukaryotic tubulin. FtsZ marks the division site by forming a 'Z-ring' in association with early-stage divisomal proteins such as FtsA, ZapA, SepF, and EzrA in Gram-positive bacteria (*Gamba et al., 2009*; *Lutkenhaus et al., 2012*). Late-stage divisomal proteins such as DivIVA, FtsL, DivIB, and various penicillin-binding proteins (PBPs) subsequently assemble to carry out cell division and facilitate the separation and creation of identical daughter cells.

GpsB is a DivIVA-like protein that is highly conserved in Firmicutes (*Halbedel and Lewis, 2019*; *Hammond et al., 2019*). While GpsB is dispensable, or conditionally essential in most Firmicutes (*Rismondo et al., 2016*; *Fleurie et al., 2014*; *Claessen et al., 2008*; *Tavares et al., 2008*), it was reported to be essential for growth in *Staphylococcus aureus* (*Santiago et al., 2015*; *Gillaspy, 2006*). However, recent studies indicate that *gpsB* may not be essential, but nevertheless plays a significant role in maintaining *S. aureus* cell morphology (*Sutton et al., 2023*; *Bartlett et al., 2023*; *Costa et al., 2023*). In particular, the importance of *S. aureus* GpsB (*Sa* GpsB) for cell division is underscored by its unique ability to regulate FtsZ polymerization in *S. aureus* (*Eswara et al., 2018*). At this time, GpsB-FtsZ interaction has not been reported in other bacteria. *Sa* GpsB has also been reported to interact with other cell division proteins such as EzrA (*Steele et al., 2011*), DivIVA (*Bottomley et al., 2017*), and the wall teichoic acids (WTA) biosynthesis/export proteins TarO and TarG (*Kent, 2024*; *Hammond et al., 2022*).

The interaction of GpsB with PBPs has also been investigated in several bacteria, including *Listeria monocytogenes* (*Lm*) PBPA1, *Streptococcus pneumoniae* (*Sp*) PBP2a, and *B. subtilis* (*Bs*) PBP1 (*Halbedel and Lewis, 2019*). These studies show that GpsB binds to a short, ~5- to 30-residue N-terminal sequence found on the cytosolic region of these bitopic PBPs, referred to as an N-terminal 'mini-domain' (*Rismondo et al., 2016*; *Cleverley et al., 2019*). Cleverley et al. found PBP mini-domains containing an (S/T)-R-X-X-R-(R/K) motif directly interact with the N-terminal domain of GpsB by forming electrostatic interactions and hydrogen bonds with a shallow, acidic cavity located at the GpsB dimer interface (*Figure 1—figure supplement 1A–C*) *Cleverley et al., 2019*.

While *Lm*, *Sp*, and *Bs* have at least six annotated PBPs (*Kocaoglu et al., 2015*; *Korsak et al., 2010*; *Kunst et al., 1997*), there are only four in *S. aureus*: PBP1, PBP2, PBP3, and PBP4 (*Gillaspy, 2006*) - five in methicillin-resistant *S. aureus* (MRSA) which expresses an additional β-lactam insensitive PBP, PBP2a (*Hartman and Tomasz, 1984*). *S. aureus* is further distinguished by its highly cross-linked peptidoglycan and can readily become resistant to β-lactams via PBP4 and the acquisition of PBP2a (*Łeski and Tomasz, 2005*; *Snowden and Perkins, 1990*). It is believed this β-lactam-resistant phenotype relies on WTA assembly, which influences the function and localization of PBP4 and PBP2a (*Atilano et al., 2010*). While PBP2a is a historically recognized element of antibacterial resistance, PBP4 has recently been found to contribute to β-lactam insensitivity (*Chatterjee et al., 2017*). As the sole class C PBP in *S. aureus,* PBP4 bears the fold and architecture of a carboxypeptidase, but uniquely catalyzes both transpeptidase and carboxypeptidase reactions (*Finan et al., 2001*; *Henze and Berger-Bächi, 1995*; *Navratna et al., 2010*; *Wyke et al., 1981*).

In this report, we show that *Sa* GpsB directly binds to FtsZ and PBP4 through a signature GpsB recognition sequence. Further analysis of the GpsB N-terminal domain reveals unique conformations and innate flexibility that is integral to the function of GpsB. Together, these findings provide insight into the unique role of GpsB in synchronizing FtsZ dynamics with cell wall synthesis during cell division in *S. aureus*.

## Results

### The crystal structure of *Sa* GpsB N-terminal domain reveals an atypical asymmetric dimer

The full-length *Sa* GpsB is a relatively small protein of 114 residues. Its N-terminal domain homodimerizes as a coiled-coil, while its smaller C-terminal domain homotrimerizes, forming a hexamer as the biological unit (*Halbedel and Lewis, 2019*; *Rismondo et al., 2016*; *Cleverley et al., 2016*). Using X-ray crystallography, we solved the structure of the N-terminal domain (residues 1–70) of *Sa* GpsB at 1.95 Å resolution in the P2$_1$ space group (*Figure 1A*; *Supplementary file 1*), with four monomers per asymmetric unit forming two dimers (dimers A and B). The overall structure of GpsB is similar to

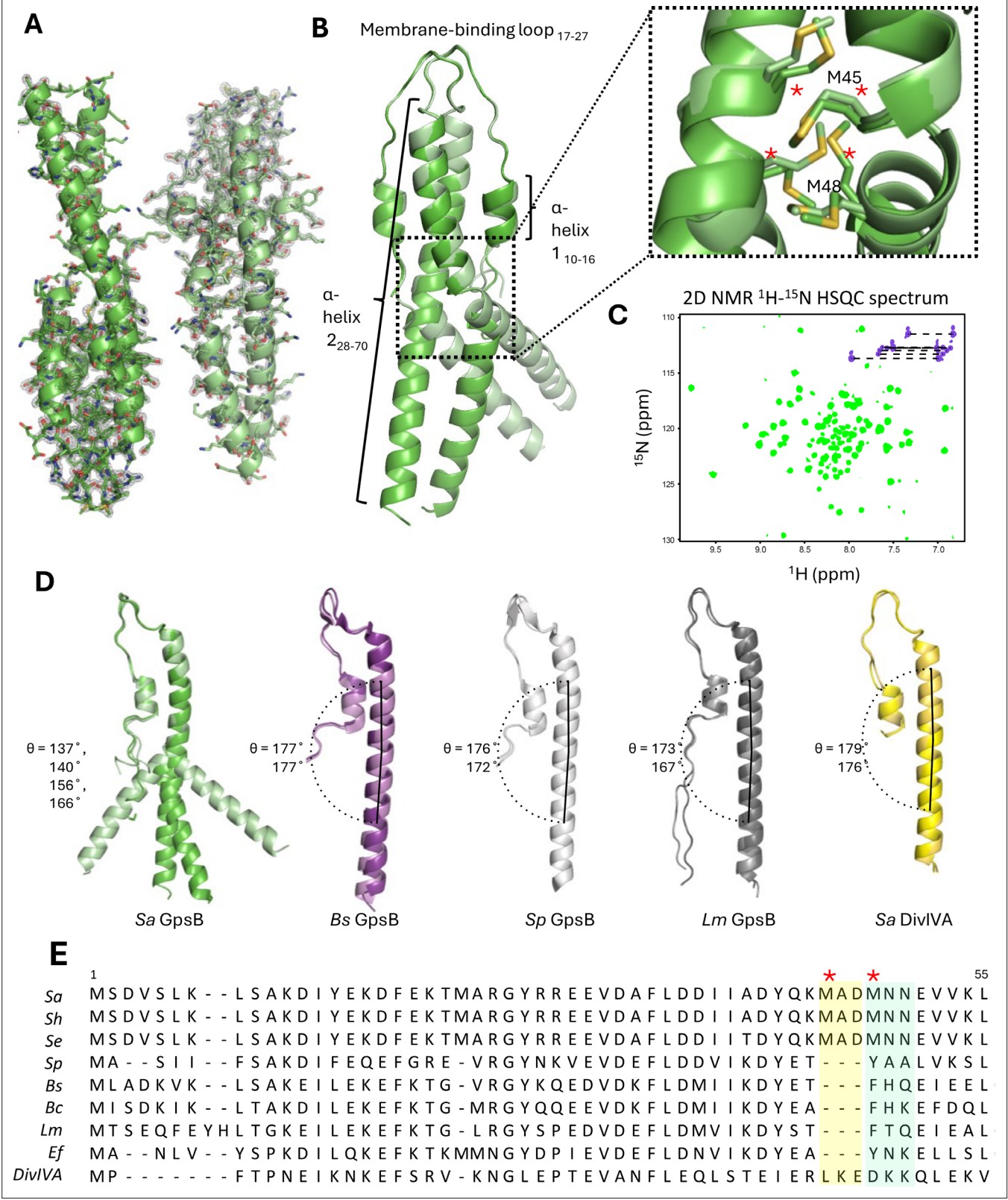

**Figure 1.** Crystal structure of the N-terminal domain of *Sa* GpsB. (**A**) Two dimers lie antiparallel in the asymmetric unit from the P2₁ space group. The 2F$_o$-F$_c$ electron density map, shown in gray, is contoured at 1σ with a resolution of 1.95 Å. (**B**) Superimposition of GpsB dimers reveals the dimers splay at a hinge region, formed by a cluster of four interlocked Met sidechains. *designates position in multisequence alignment shown in panel E. (**C**) The ¹H-¹⁵N HSQC spectrum of the GpsB N-terminal domain, *Sa* GpsB^WT₁₋₇₀, shows a significantly higher number of signals compared to what is expected of

*Figure 1 continued on next page*

*Figure 1 continued*

a symmetric dimer based on the sequence of the domain, indicating the presence of conformational heterogeneity. Backbone and Asn/Gln sidechain amide signals are shown in green and purple respectively. (**D**) Comparison of different GpsB/DivIVA monomers from previously solved structures. Pitch angles were determined by placing a marker atom at the centroid of the beginning, midpoint 'hinge', and end of each helix, then measuring the angle. (**E**) Multisequence alignment of GpsB within select members of the Firmicutes phylum and with *S. aureus* DivIVA. Staphylococci GpsB contain a three-residue insertion that forms the hinge region, either MAD or MNN, depending on the sequence alignment parameters. The two Met residues (four per dimer) of the hinge region are designated with a red *. *Sa - S. aureus, Sh - S. haemolyticus, Se - S. epidermidis, Sp - S. pneumoniae, Bs - B. subtilis, Bc - B. cereus, Lm - L. monocytogenes, Ef - E. faecalis, DivIVA - S. aureus* DivIVA.

The online version of this article includes the following figure supplement(s) for figure 1:

**Figure supplement 1.** Crystal structures of penicillin-binding protein (PBP) mini-domains in complex with the N-terminal domain of their cognate GpsB, published by *Cleverley et al., 2019*.

**Figure supplement 2.** ¹H-¹⁵N HSQC HSQC identifies conformational heterogeneity within the N-terminal domain of *Sa* GpsB.

**Figure supplement 3.** Molecular dynamics simulations and homology models of GpsB mutants reveal potential structural origins of GpsB stability and flexibility.

previously determined GpsB orthologs (*Figure 1—figure supplement 1*; *Rismondo et al., 2016*; *Cleverley et al., 2019*) and retains the same fold of DivIVA (*Oliva et al., 2010*), though it shares much less sequence similarity (*Figure 1E*). Dimerization of the N-terminal domain is facilitated by a pattern of nonpolar residues every three to four residues, promoting the formation of a hydrophobic core in the coiled-coil. This N-terminal domain is partitioned into three regions: a relatively short α-helix with approximately two turns (residues 10–16), a second longer α-helix (residues 28–70) that forms a coiled-coil, and an amphipathic 10-residue loop (residues 17–27) that links these two helices and intertwines with the adjacent protomer, 'capping' GpsB (*Figure 1B*). This loop region is proposed to interact with the inner leaflet of the cell membrane (*Halbedel and Lewis, 2019*).

The most unique structural feature of *Sa* GpsB is a hinge forming at the midpoint of its N-terminal domain that causes each protomer to bend, adopting pitch angles (θ) of 137–140° (dimer A) and 156–166° (dimer B). In contrast, GpsB protomers from *Lm, Bs, Sp*, and *Sa* DivIVA are almost linear (θ=167–179°) and practically indistinguishable when superimposed (*Figure 1D*). We find that there is a 3-amino acid insertion in the *Sa* GpsB sequence where the helicity is disrupted and that a cluster of four Met residues (Met45, Met48) are interlocked at the dimer interface at this position (*Figure 1B*; dotted rectangle). These Met residues are only found in *Staphylococci* (*Figure 1E*) and take the place of an aromatic Tyr or Phe present in other orthologs, which normally stabilize the core via π stacking interactions. While the 3-amino acid insertion likely disrupts the continuity of the coiled-coil, methionine is one of the most flexible aliphatic amino acids and may further contribute to the conformational flexibility.

Experiments using solution state NMR spectroscopy further support the notion of intrinsic conformational heterogeneity, suggesting it is a bona fide structural feature rather than a crystallization artifact. The *Sa* GpsB$^{WT}_{1-70}$ construct contains a two-residue N-terminal extension (GH) after removal of the purification tag, and thus a total of 72 signals are expected. The 2D ¹H-¹⁵N HSQC spectrum shows good signal dispersion (*Figure 1C*), which is characteristic of a folded domain. However, a total of 100 well-resolved backbone amide signals are observed, and seven pairs of signals are detected in the Asn/Gln sidechain region instead of four pairs (*Figure 1C*, *Figure 1—figure supplement 2B*). The appearance of the additional signals is not due to proline *cis/trans* isomerization since there are no prolines in the sequence, nor is it due to self-aggregation of the dimers to form a dimer-of-dimers or higher order oligomers, because raising the concentration from 160 to 700 μM has no effect on the number of signals in the spectrum (*Figure 1—figure supplement 2B*). In addition, the ¹H-¹⁵N HSQC of *Sa* GpsB$^{WT}_{1-70}$ exhibits differential linewidths, with a set of 13 narrow signals at the center of the spectrum where disordered segments appear, and a larger set of 87 signals dispersed throughout the spectrum. The number of the narrow signals corresponds well with the number of residues found at the flexible N-terminus, suggesting that the appearance of extra signals is due to sampling of multiple conformations of the coiled-coil. Furthermore, three of the four Asn/Gln sidechain pairs showing chemical shift degeneracy are adjacent to the hinge (*Figure 1—figure supplement 2A*) providing strong evidence that the observed conformational heterogeneity occurs at this region.

Molecular dynamics (MD) simulations using Gromacs v. 5.0.4 and a CHARMM36m force field also support the idea that conformational flexibility exists in solution. After approximately 100 ns, dimer

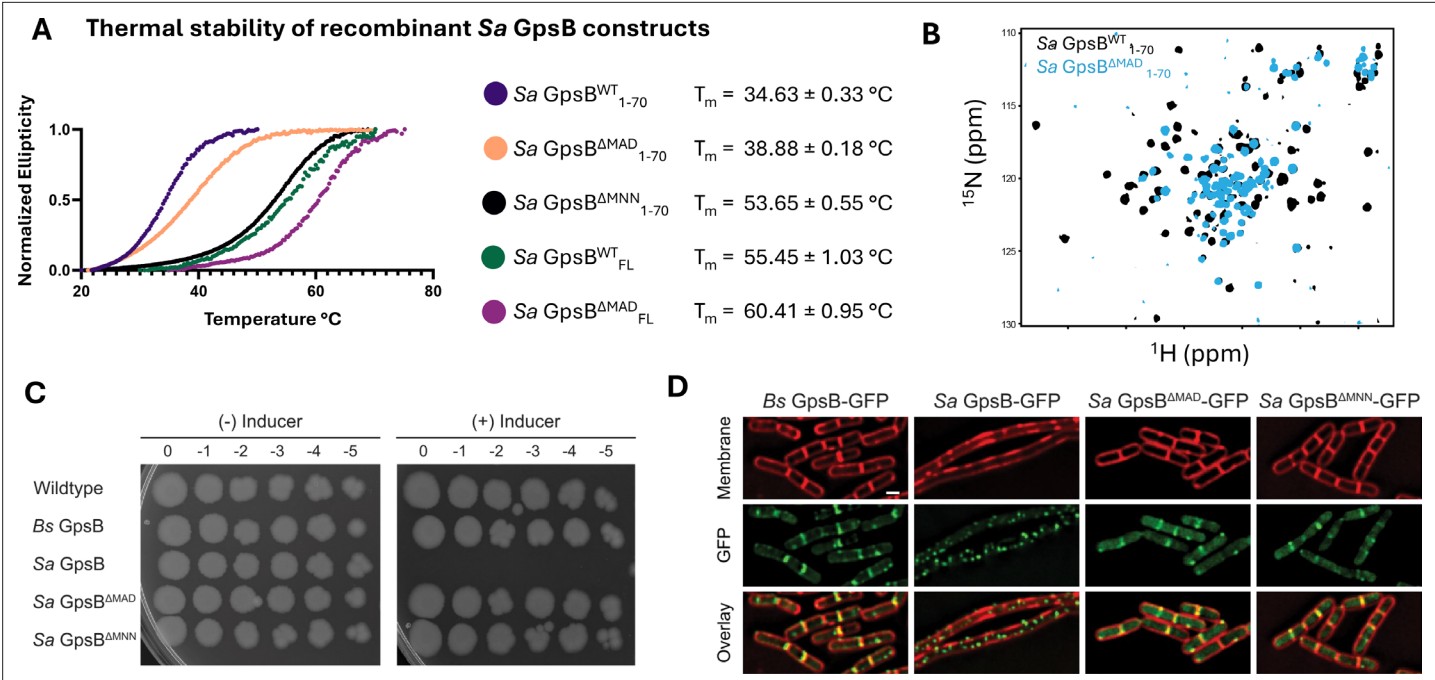

**Figure 2.** Deletion of a three-residue insertion in *Sa* GpsB increases thermal stability in solution and abolishes toxicity in *B. subtilis*. (**A**) Circular dichroism (CD) melt profiles of recombinantly expressed *Sa* GpsB constructs reveal ΔMAD and ΔMNN mutants have increased $T_m$, compared to wildtype (WT) *Sa* GpsB. (**B**) Overlays of the $^1$H-$^{15}$N HSQC of spectrum of *Sa* GpsB$^{WT}$ $_{1-70}$ and Sa GpsB$^{ΔMAD}$ $_{1-70}$ suggest there are significant differences in the conformational properties of these two proteins. (**C**) Serial dilutions of *B. subtilis* strains harboring inducible *Bs* GpsB (GG18), *Sa* GpsB (GG7), *Sa* GpsB ΔMAD (LH119), and *Sa* GpsB ΔMNN (LH115), plated on LB plates without (left) and with (right) 1 mM IPTG demonstrate WT *Sa* GpsB is lethal, but ΔMAD *Sa* GpsB and ΔMNN *Sa* GpsB are not. (**D**) Fluorescence micrographs showing the protein localization of *Bs* GpsB-GFP (GG19), *Sa* GpsB-GFP (GG8), *Sa* GpsB-GFP ΔMAD (LH126), and *Sa* GpsB-GFP ΔMNN (LH116). Cell membrane was visualized using SynaptoRed membrane dye (1 µg/mL). Scale bar, 1 µm. In contrast to WT *Sa* GpsB, strains of *B. subtilis* that overexpress ΔMAD and ΔMNN *Sa* GpsB have similar cellular morphology to WT *B. subtilis* and these proteins localize to the division septum.

The online version of this article includes the following source data and figure supplement(s) for figure 2:

**Figure supplement 1.** Deletion of MNN residues more drastically affects the function of *Sa* GpsB than deletion of MAD residues.

**Figure supplement 1—source data 1.** Images of total protein gel and anti-GpsB western blot.

**Figure supplement 2.** Deletion of MAD/MNN residues renders *Sa* GpsB less functional, but only minimally affects the function of heterocomplex with unmutated counterpart.

**Figure supplement 2—source data 1.** Images of total protein gel and anti-GpsB western blot.

1 adopts an ~180° pitch angle that occurs concomitantly to major fluctuations in dimer 2 in a separate simulation (*Figure 1—figure supplement 3A*), although the preference for continuous helical structures (corresponding to ~180° pitch angles) may sometimes be influenced by specific force field parameters.

## A three-residue insertion unique to *Staphylococci* GpsB disrupts the coiled-coil pattern and destabilizes the structure of *Sa* GpsB

A unique element of *Sa* GpsB that likely contributes to its distinct conformational flexibility are three extra residues located at the hinge region - MAD or MNN (depending on the alignment parameters; *Figure 1E*), roughly corresponding to an extra turn in the helix. Deleting either MAD or MNN produces a homology model where residues 45–70, normally displaced by one turn, are now aligned with their orthologous residue pair (*Figure 1—figure supplement 3B*). To investigate the relative degree of stability imparted by these residues, GpsB mutant lacking the extra MAD or MNN residues at the hinge region was recombinantly constructed and purified, and $T_m$ (melting temperature) was determined using circular dichroism (CD) spectroscopy (*Figure 2A*). This experiment shows the ΔMAD and ΔMNN GpsB have superior thermal stability to wildtype (WT) *Sa* GpsB for both the N-terminal

domain (1–70) and full-length constructs. Perhaps the most notable finding from this experiment was that, while the $T_m$ of *Sa* GpsB$^{\Delta MAD}_{1-70}$ (38.88 ± 0.18°C) was only modestly higher than *Sa* GpsB$^{WT}_{1-70}$ (34.63 ± 0.33°C), the $T_m$ of *Sa* GpsB$^{\Delta MNN}_{1-70}$ (53.65 ± 0.55°C) was highly stable, approximately 1.5-fold higher than that of *Sa* GpsB$^{WT}_{1-70}$ and comparable to the thermal stability of the full-length *Sa* GpsB-$^{WT}_{FL}$. Though the full-length *Sa* GpsB$^{\Delta MNN}_{FL}$ was not analyzed, we expect that its $T_m$ would be higher than both *Sa* GpsB$^{WT}_{FL}$ and *Sa* GpsB$^{\Delta MAD}_{FL}$ based on the experiments assessing the 1–70 constructs.

The conformational properties of the *Sa* GpsB$^{\Delta MAD}_{1-70}$ mutant were also different when analyzed by $^1$H-$^{15}$N HSQC (*Figure 2B*). Most of the dispersed signals from the coiled-coil region experience severe broadening, in many cases beyond detection, as well as changes in chemical shift, while all narrow signals at the center of the spectrum show no change in linewidth or positions. Signal broadening is caused by changes in the rate of interconversion between the available conformations from the intermediate-slow for WT (s) to the intermediate-fast exchange regime for ΔMAD (ms), but without altering the overall number of states, as seven pairs of Asn/Gln sidechain pairs of signals are observed. Conformational rigidity is a well-established correlate of thermal stability (*Karshikoff et al., 2015*), and may contribute to the enhanced $T_m$ of *Sa* GpsB$^{\Delta MNN}$ and *Sa* GpsB$^{\Delta MAD}$. The aforementioned homology model (*Figure 1—figure supplement 3B*) corresponding to *Sa* GpsB$^{\Delta MNN}$ and *Sa* GpsB$^{\Delta MAD}$ reveals several residues potentially form stronger interactions than the WT. These include an intra-helical electrostatic interaction that replaces a potential repulsion between Asp47 and Glu51 with Asn47/Lys51 in *Sa* GpsB$^{\Delta MAD}$ and Asp47/Lys51 in *Sa* GpsB$^{\Delta MNN}$ (*Figure 1—figure supplement 3C*). The favorable Asp47/Lys51 electrostatic interactions in GpsB$^{\Delta MNN}$ may further contribute to its higher thermostability.

## The MAD/MNN insertion is critical for GpsB function

Previously, we reported that overproduction of *Sa* GpsB in *B. subtilis* causes cell division arrest which eventually leads to filamentation and cell lysis (*Eswara et al., 2018*). We used this system to probe the significance of the flexibility provided by MAD/MNN residues for the function of GpsB. First, we conducted a growth assay on solid medium by spotting serial dilutions of cells of *B. subtilis* WT and cells harboring an inducible copy of *Bs gpsB*, *Sa gpsB*, *Sa gpsB$^{\Delta MAD}$*, or *Sa gpsB$^{\Delta MNN}$*. As previously established, overproduction of *Bs* GpsB is not lethal, but *Sa* GpsB is (*Figure 2C*; *Eswara et al., 2018*; *Hammond et al., 2022*). In contrast, overproduction of either ΔMAD or ΔMNN *Sa* GpsB is not lethal and cells grow as well as the negative controls (*B. subtilis* WT and inducible *Bs* GpsB strain). These results suggest the hinge region of *Sa* GpsB, characterized by two Met residues (*Figure 1B*), is essential for the normal function of *Sa* GpsB as assessed by lethal phenotype in *B. subtilis*. To further investigate the impact of these mutations, we examined GFP tagged ΔMAD and ΔMNN *Sa* GpsB in *B. subtilis* using high-resolution fluorescence microscopy. As reported previously, and as shown in *Figure 2D*, *Bs* GpsB-GFP localizes to the division site (*Claessen et al., 2008*; *Tavares et al., 2008*; *Hammond et al., 2022*) and does not lead to filamentation upon overproduction (2.09 ±0.49 μm; n=50), but *Sa* GpsB-GFP forms foci throughout the entire cell, and causes severe filamentation (22.83 ±16.51 μm; n=21) (*Eswara et al., 2018*). However, ΔMAD and ΔMNN variants of *Sa* GpsB-GFP do not cause filamentation (2.13 ±0.44 μm and 2.10 ±0.43 μm respectively; n=50) and are clearly localized at the division site. It has been noted previously that *Sa* GpsB-GFP also localizes to the division sites at initial stages prior to causing filamentation at a lower inducer concentration (*Eswara et al., 2018*). Taken together, this data suggests the ΔMAD and ΔMNN mutants presumably interact with the *B. subtilis* cell division machinery to allow for division site localization, but fail to elicit lethal filamentation and toxicity (*Eswara et al., 2018*).

We also investigated the phenotypes of *S. aureus* strains that overexpress *Sa* GpsB$^{\Delta MAD}$ and *Sa* GpsB$^{\Delta MNN}$. First, we conducted a growth assay by spotting serial dilutions of *S. aureus* strains harboring an empty vector (EV) or an additional plasmid-based inducible copy of *Sa gpsB*, *Sa gpsB$^{\Delta MAD}$*, or *Sa gpsB$^{\Delta MNN}$*. Overproduction of *Sa* GpsB leads to a 100- to 1000-fold growth inhibition compared to EV control (*Figure 2—figure supplement 1A*). Interestingly, while cells overproducing *Sa* GpsB$^{\Delta MNN}$ grew similarly to the EV control, *Sa gpsB$^{\Delta MAD}$* overexpression resulted in growth inhibition comparable to *Sa* GpsB. We hypothesized that the difference in phenotype is due to differential affinity of native *Sa* GpsB (produced from chromosomal locus) to *Sa* GpsB$^{\Delta MAD}$ and *Sa* GpsB$^{\Delta MNN}$ mutants (produced from a plasmid-based system), leading to varying propensity for homo/hetero complex formation. To test this, we conducted bacterial two-hybrid analysis in MacConkey plates and liquid culture as reported

previously (**Hammond et al., 2022**; **Battesti and Bouveret, 2012**). As shown in **Figure 2—figure supplement 1B**, the affinity of *Sa* GpsB$^{ΔMNN}$ to itself appears to be greater compared to *Sa* GpsB$^{ΔMNN}$ and *Sa* GpsB. *Sa* GpsB$^{ΔMAD}$ appears to have similar affinity to itself and *Sa* GpsB, but it is weaker compared to the self-interaction of *Sa* GpsB. Thus, the differential affinity between mutant *Sa* GpsB and WT *Sa* GpsB could underlie the observed difference in phenotypes. Next, we analyzed the cell morphology of *S. aureus* strains overproducing ΔMAD/ΔMNN mutants. As we have previously shown, overproduction of *Sa* GpsB leads to cell size enlargement due to cell division inhibition (**Eswara et al., 2018**). Using fluorescence microscopy (**Figure 2—figure supplement 1C and D**), we re-confirmed the increase in cell diameter in cells overproducing *Sa* GpsB (1.00 ±0.19 µm) when compared to the EV control (0.90 ±0.13 µm). In agreement with the lethal plate phenotype (**Figure 2—figure supplement 1A**), cells overproducing *Sa* GpsB$^{ΔMAD}$ also displayed a statistically significant increase in cell diameter (0.95 ±0.16 µm), while *Sa* GpsB$^{ΔMNN}$ did not (0.91 ±0.15 µm) and resembled EV control. We also ensured the stable production of *Sa* GpsB, ΔMAD, and ΔMNN via western blotting (**Figure 2—figure supplement 1E**). Lastly, by GFP tagging, we observed that both mutants localize to the division site similar to *Sa* GpsB-GFP (**Figure 2—figure supplement 1F**; **Eswara et al., 2018**; **Hammond et al., 2022**). In summary, ΔMNN is less functional compared to ΔMAD in terms of causing growth inhibition on plate and cell enlargement phenotype, however both ΔMAD and ΔMNN mutants are able to localize to division sites.

To further test the functionality of the hinge region mutants, we generated a *Sa gpsB* knockout strain using CRISPR/Cas9-based approach (**Chen et al., 2017**). As the sole copy, *Sa gpsB*$^{ΔMAD}$ appears to be less toxic compared to the inducible WT *Sa gpsB*, suggesting that it is partially functional (**Figure 2—figure supplement 2A**). However, the ability of *Sa gpsB*$^{ΔMNN}$ to impact *S. aureus* growth is severely impaired and resembles the EV control. This observation was consistent with the cell diameter quantification of different strains (**Figure 2—figure supplement 2B**). We also ensured the stability of these mutants is unaffected via western blotting (**Figure 2—figure supplement 2C**). To probe whether these mutants inhibit the native function of *Sa* GpsB when in a heteromer, similar to other previously isolated and functionally defective *Sa* GpsB mutants (such as *Sa gpsB*$^{ΔLEE}$) (**Hammond et al., 2022**), we co-expressed *Sa gpsB-gfp* and *Sa gpsB*, *Sa gpsB*$^{ΔMAD}$, or *Sa gpsB*$^{ΔMNN}$ in *B. subtilis*. We observed that co-expression of hinge region mutants does not completely abolish the toxicity of *Sa gpsB-gfp* (**Figure 2—figure supplement 2D**). This result suggests that, although the *Sa gpsB*$^{ΔMAD}$ and *Sa gpsB*$^{ΔMNN}$ mutants are by themselves nonfunctional in *B. subtilis* (**Figure 2C**), they do not drastically inhibit the function of WT GpsB in the heteromer. We hypothesize that the deletion mutants lacking the flexible hinge region may to some degree potentially mimic certain GpsB conformations with varied activities. Thus the new data may indicate that the protomers adopting different conformations within the GpsB complex can function independently of one another, providing a mechanism to fine-tune the function of the whole complex.

## The C-terminus of *Sa* FtsZ binds to the N-terminal domain of *Sa* GpsB through a conserved (S/T/N)-R-X-X-R-(R/K) motif

One of the few proteins known to interact with *Sa* GpsB is the tubulin-like GTPase, FtsZ - a central cell division protein that marks the division site in nearly all bacteria. We previously demonstrated that GpsB directly interacts with *Sa* FtsZ to stimulate its GTPase activity and modulate its polymerization characteristics (**Eswara et al., 2018**). However, the molecular basis for this interaction was not known. Remarkably, we discovered that the last 12 residues of *Sa* FtsZ (N-R-E-E-R-R—S-R-R-T-R-R), also known as the C-terminal variable (CTV) region (**Buske and Levin, 2012**), are a repeated match of the consensus GpsB-binding motif (S/T-R-X-X-R-(R/K)) found in the N-termini of *Bs* PBP1, *Lm* PBPA1, and *Sp* PBP2a. In this instance, the first motif bears an Asn instead of a Ser/Thr. Given the similar physicochemical properties of Asn as a small polar amino acid, it can likely replace Ser or Thr without any functional significance. To our knowledge, the interaction between GpsB and FtsZ is unique to *S. aureus* (**Hammond et al., 2019**), which is consistent with the absence of this motif in FtsZ orthologs from other organisms (**Figure 3A**).

To initially test whether the predicted FtsZ GpsB-binding motif directly binds to *Sa* GpsB, we purified and titrated the terminal 66 residues of *Sa* FtsZ (325–390) against full-length *Sa* GpsB using surface plasmon resonance (SPR), revealing a dose-dependent interaction ($K_D$ = 40.21 ± 1.77 µM; **Figure 3B**). A simultaneous titration against only the *Sa* GpsB N-terminal domain (1–70) demonstrates

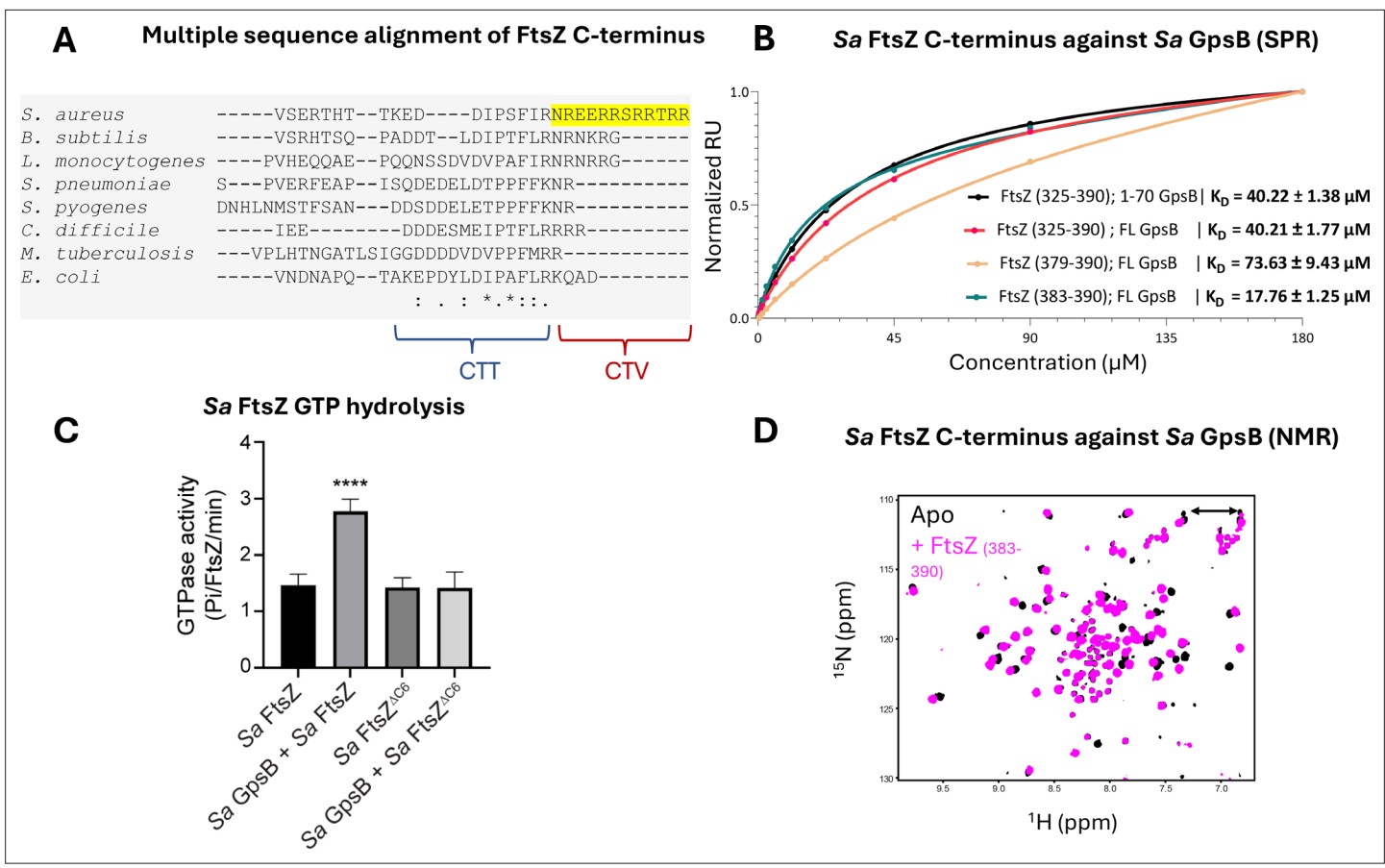

**Figure 3.** *Sa* FtsZ contains a repeated GpsB recognition motif at its C-terminus. (**A**) A multiple sequence alignment of the FtsZ C-terminus from different representative bacteria reveals that there is a repeated GpsB recognition motif in *Sa* FtsZ (highlighted region) and that it is unique to this bacterium. (**B**) Surface plasmon resonance (SPR) titration of peptides corresponding to several segments of *Sa* FtsZ against *Sa* GpsB (residues 1–70 or full length). A titration of *Sa* FtsZ (residues 325–390) against 1–70 GpsB (black), corresponding to the N-terminal domain, shows binding can be isolated to this region. (**C**) When incubated with *Sa* GpsB, a *Sa* FtsZ mutant with a C-terminal truncation (SRRTRR, FtsZ$^{\Delta C6}$) has significantly lower GTP hydrolysis compared to its full-length counterpart. GTP hydrolysis was measured by monitoring inorganic phosphate (P$_i$) released (µmoles/min) by either FtsZ or FtsZ$^{\Delta C6}$ (30 µM) in the absence and presence of GpsB (10 µM). The plot is the average of n=6 independent data sets. p-Value for **** is <0.0001. (**D**) Overlays of the ${}^1$H-${}^{15}$N HSQC of spectrum of *Sa* GpsB (1–70) in the absence (black) and in the presence of FtsZ (383–390; pink). The boxed region highlights the only sidechain pair of signals that becomes affected by the addition of Ftsz. Based on a model derived from the structure of *Bs* GpsB in complex with a penicillin-binding protein (PBP)-derived peptide (*Figure 1—figure supplement 2*), it is tentatively assigned to Q43 of *Sa* GpsB.

that the C-terminus of *Sa* FtsZ binds exclusively to this domain (K$_D$ = 40.22 ± 1.38 µM). This is further supported by assays with *Sa* FtsZ CTV (379–390; K$_D$ = 73.63 ± 9.43 µM) and the final eight residues of *Sa* FtsZ (383–390; K$_D$ 17.76±1.25 µM). The described biophysical affinity aligns with previous cellular studies (*Cleverley et al., 2019*), and is consistent with our hypothesis that the impetus for binding is the (S/T/N)-R-X-X-R-(R/K) motif which associates with the PBP-binding pocket (*Figure 1—figure supplement 1*). Unlike the previously identified GpsB-binding motifs located at the N-termini of PBPs, this is the first time such a motif has been found at the C-terminus of a GpsB-binding protein. The approximate fourfold difference in affinity between the CTV and the final eight residues could be a result of the inclusion of two Glu residues in the CTV (E381, E382), which may experience repulsion from the negatively charged PBP-binding pocket of GpsB. Notably, the GpsB recognition motif of *Sa* FtsZ is adjacent to the C-terminus carboxylate, which imparts an additional negative charge.

To further characterize the interaction between *Sa* FtsZ and *Sa* GpsB, we conducted a GTPase assay with *Sa* GpsB and *Sa* FtsZ or *Sa* FtsZ with the C-terminal six residues truncated (*Sa* FtsZ$^{\Delta C6}$). Briefly, in our previous report (*Eswara et al., 2018*), we found that *Sa* GpsB enhances the GTPase activity of *Sa* FtsZ. Therefore, we hypothesized that truncation of the terminal six residues of *Sa* FtsZ would eliminate the GpsB-mediated enhancement of GTPase activity. As shown in *Figure 3C*, and as

reported previously, addition of *Sa* GpsB enhanced the GTPase activity of *Sa* FtsZ. However, this effect was not seen in *Sa* FtsZ$^{\Delta C6}$, suggesting that the last six C-terminal residues of *Sa* FtsZ is likely where the interaction with *Sa* GpsB occurs.

The interaction of the *Sa* GpsB N-terminal domain with the *Sa* FtsZ derived peptide (383–390) was also monitored using NMR (*Figure 3D*). Addition of the octapeptide to $^{15}$N-labeled N-terminal domain results in a significant chemical shift perturbation to a small number of dispersed signals in the $^{1}$H-$^{15}$N HSQC spectrum, suggesting that the interaction is highly localized and occurs through the coiled-coil region without disturbing the conformational heterogeneity of the dimer. The available structures of *Bs* GpsB and *Sp* GpsB in complex with PBP-derived peptides (*Figure 1—figure supplement 1A–C*) suggest the complex is stabilized through interactions with helices 1 and 2, as well as part of the h1-h2 capping loop (*Cleverley et al., 2019*). The N-terminal Arg of *Sa* FtsZ octapeptide utilized in our experiments is expected to be placed in the PBP-binding pocket near Q43, based on the complex structures of GpsB homologs. Indeed, only one of the Asn/Gln pairs of signals in the $^{1}$H-$^{15}$N HSQC is shifted upon addition of the peptide, while in agreement with our model all other sidechain signals lie far from the binding site (*Figure 1—figure supplement 2A and B*), suggesting that *Sa* GpsB recognizes partner proteins in a conserved manner.

## The cytosolic, C-terminal mini-domain of *Sa* PBP4 binds to *Sa* GpsB through its (S/T/N)-R-X-X-R-(R/K) recognition motif

*S. aureus* has an unusually low number of PBPs in its genome (*Gillaspy, 2006*). Although PBP1, PBP2, PBP2a, and PBP3 all have a cytoplasmic N-terminal 'mini-domain', none bear a GpsB recognition motif (*Figure 4A*). Using SPR, we confirmed their cytoplasmic mini-domains have no affinity for *Sa* GpsB, which is in agreement with the findings from previous bacterial two-hybrid assays (*Steele et al., 2011*). Remarkably, PBP4, the only class C PBP encoded in the *S. aureus* genome, which purportedly functions as both a carboxypeptidase and transpeptidase PBP (*Maya-Martinez et al., 2018*), has a short, cytosolic C-terminal mini-domain with the sequence of N-R-L-F-R-K-R-K, satisfying the consensus GpsB-binding motif (S/T/N)-R-X-X-R-(R/K) found in *S. aureus* FtsZ and in orthologous PBPs found to bind to GpsB. Next, using SPR, we demonstrate this *Sa* PBP4 C-terminal octapeptide binds to *Sa* GpsB with a K$_D$ of 48.61±1.86 μM (*Figure 4A*). Supporting this finding is a potential interaction between *Sa* PBP4 and *Sa* GpsB previously noted in a bacterial two-hybrid study (*Kent, 2024*).

Initial efforts to obtain a crystal structure of *Sa* PBP4 and *Sa* FtsZ with *Sa* GpsB were unsuccessful. A key factor preventing the formation of this complex were the tight interactions forming at the crystal packing interface. By extending the asymmetric unit, we found each GpsB dimer coordinates two others in a head-to-head arrangement (*Figure 4—figure supplement 1A and B*). This involves the insertion of Arg and Lys residues from the membrane-binding loop into the PBP-binding site from the adjacent GpsB protomer, mimicking the binding mode observed between PBP mini-domains and GpsB in orthologous structures, such as *Bs* PBP1 (*Figure 4—figure supplement 1C and D*; *Cleverley et al., 2019*). It is unclear whether this head-to-head interaction occurs in the cell or is simply a crystallization artifact. Nonetheless, it is apparent this interaction would need to be disrupted to capture interactions with a binding partner. To do so, we generated an R24A mutant because of its central role in the head-to-head interaction and its distance from the PBP-binding groove. This point mutation successfully disrupted the occluded crystal packing interface and allowed us to determine a 2.40 Å resolution structure of *Sa* PBP4 peptide bound to *Sa* GpsB (*Figure 4B*, *Supplementary file 1*). Unambiguous electron density for the *Sa* PBP4 C-terminal octapeptide, which adopted an α-helix, was resolved at the PBP-binding groove of GpsB. This structure clearly shows that two of these residues, Arg425 and Arg428, are key components of the *Sa* PBP4-*Sa* GpsB interaction, where they form multiple hydrogen bonds with the main chain amides of *Sa* GpsB Ile13, Tyr14, Lys16, and the sidechain hydroxyl of Tyr26. Furthermore, Arg425 and Arg428 form two salt bridges with the carboxyl sidechain of Asp32. Additionally, *Sa* GpsB Asp36 appears to play a major role in stabilizing the PBP4 α-helix by forming two hydrogen bonds with the backbone nitrogen of Arg425 and Leu426, an arrangement that is only possible when these two residues are part of an α-helix. Overall, the complex of *Sa* PBP4/*Sa* GpsB is very similar to *Bs* PBP1/*Bs* GpsB (*Figure 1—figure supplement 1*, *Figure 4—figure supplement 1C and D*) and is distinguished by the interactions of two Arg residues with the backbone and acidic sidechains lining the PBP-binding groove. We further used AlphaFold2-Multimer to study the interactions between *Sa* GpsB and FtsZ (*Evans et al., 2021*). Although attempts using full-length

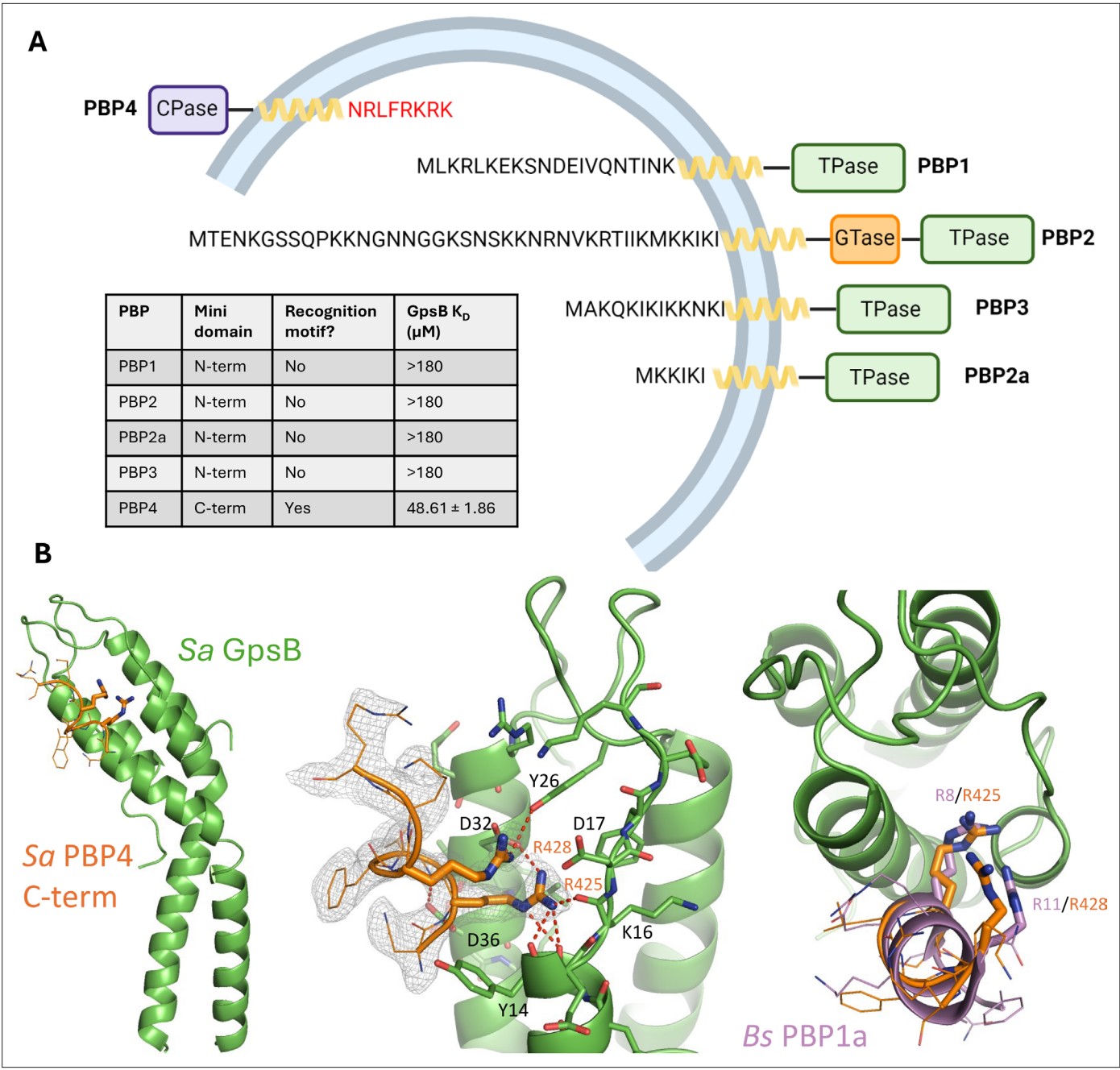

**Figure 4.** The C-terminal mini-domain of PBP4 directly interacts with GpsB. (**A**) Domain representation of the four/five *S. aureus* (COL)/methicillin-resistant *S. aureus* (MRSA) (USA300) penicillin-binding proteins (PBPs). Each protein is shown from the N-terminus (left) to the C-terminus (right). All four transpeptidase PBPs - PBP1, PBP2, PBP3, and PBP2a - lack a GpsB recognition motif on their N-terminal, cytosolic mini-domain. In contrast, the C-terminal mini-domain of PBP4, the sole *S. aureus* class C PBP, contains this motif (NRLFRKRK, red). The dissociation constants were determined with SPR (n=2). (**B**) Crystal structure of *Sa* GpsB R24A in complex with PBP4 C-terminal peptide fragment at 2.40 Å resolution. The middle panel includes the electron density map of the *Sa* PBP4 heptapeptide, $2F_o$-$F_c$=1.0σ. The right panel shows a superimposition of the *Bs* PBP1 mini-domain from the *Bs* GpsB+PBP1 complex (PDB ID 6GP7, purple) highlighting similar binding features.

The online version of this article includes the following figure supplement(s) for figure 4:

**Figure supplement 1.** The crystal packing interface of *Sa* GpsB dimers form interactions that mimic those between GpsB-PBP pairs.

**Figure supplement 2.** Surface plasmon resonance (SPR) titration of *Sa* FtsZ C-terminal peptides against *Sa* GpsB$^{ΔMAD}_{FL}$.

**Figure supplement 3.** AlphaFold2-Multimer prediction of GpsB:FtsZ$^{385-390}$ complex (cyan) superimposed onto the GpsB:PBP4 peptide crystal structure (green:orange).

GpsB and/or long segments of the FtsZ C-terminal domain did not provide satisfying results, a model with the last six residues (S-R-R-T-R-R) was obtained and the nature of interaction resembles the *Sa* GpsB complex structure with the PBP4 peptide (*Figure 4—figure supplement 3*).

## *Sa* FtsZ and *Sa* PBP4 have lower affinity for *Sa* GpsB$^{\Delta MAD}_{FL}$ compared to *Sa* GpsB$^{WT}_{FL}$

Due to the apparent importance of the three-residue insertion at the midpoint of *Sa* GpsB, we also tested the affinity of *Sa* FtsZ and *Sa* PBP4 derived peptides against *Sa* GpsB$^{\Delta MAD}_{FL}$. Using SPR we found that the affinity was significantly reduced for *Sa* GpsB$^{\Delta MAD}_{FL}$ compared to *Sa* GpsB$^{WT}_{FL}$ (*Figure 4—figure supplement 2*, *Supplementary file 2*). Dose-dependent saturation was very weak ($K_D > 200$ µM) for *Sa* PBP4 (423–431) and *Sa* FtsZ (379–390), which verged on being undetectable. Additionally, titration of $^{15}$N-labeled *Sa* GpsB$^{\Delta MAD}_{1-70}$ with the *Sa* FtsZ$_{383-390}$ peptide does not result in chemical shift changes to the GpsB$^{\Delta MAD}$ $^1$H-$^{15}$N HSQC, but only in broadening of a small number of signals, which is consistent with the higher $K_D$ measured by SPR (*Figure 1—figure supplement 2C*). While these mutants demonstrate better thermal stability (*Figure 2A*), it is possible their deletion may disrupt interactions with α-helix 1 (residues 10–16; *Figure 1B*), which may subsequently alter the size or shape of the PBP-binding pocket, thus reducing the favorable interactions that promote binding.

## Discussion

GpsB is an important protein that coordinates multiple elements of the cell wall synthesis machinery. Although GpsB is widespread among Firmicutes, it has unique structural characteristics and functional roles in *S. aureus*. In this study, we initially present the crystal structure of the *Sa* GpsB N-terminal domain (1–70) (*Figure 1A*). The characteristic coiled-coil motif demonstrates conformational flexibility resulting from a three-residue hinge region that is unique to *Staphylococci* (*Figure 1C, D, and E*). The function of this hinge region and the flexibility it imparts remains unclear, but its deletion increases thermal stability (*Figure 2A*) and weakens affinity for *Sa* FtsZ and *Sa* PBP4 (*Supplementary file 2*). Furthermore, unlike WT *Sa* GpsB, ΔMAD and ΔMNN mutants are not toxic to *B. subtilis* (*Figure 2C and D*), underscoring the cellular significance of this region. In *S. aureus*, we show that the phenotypes of ΔMAD (but not ΔMNN) mimic those of *Sa* GpsB (*Figure 2—figure supplement 1A and C*). Regardless, both ΔMAD and ΔMNN mutants localize to the division site in both *S. aureus* and *B. subtilis* suggesting they retain some level of affinity for their usual interaction partners. In *S. aureus* Δ*gpsB* background, ΔMAD and ΔMNN mutants are less functional to varying extent (*Figure 2—figure supplement 2A*). Our analysis through co-expression of ΔMAD or ΔMNN with WT *Sa* GpsB in *B. subtilis* revealed that the heterocomplex is functional (*Figure 2—figure supplement 2D*). Taken together, we believe that the hinge region in *Sa* GpsB may permit the protomers with different conformations within the complex to function independently from each other to allow for the fine-tuning of function and/or protein interactions.

Next, we identified a GpsB recognition motif on the C-termini of both *Sa* FtsZ and *Sa* PBP4, leading to biophysical, biochemical, and structural experiments providing evidence of a direct interaction between these proteins and the N-terminal domain of *Sa* GpsB (*Figure 3*). This recognition motif, which is also present, but in the N-termini of *Lm* PBPA1, *Sp* PBP2a, and *Bs* PBP1, involves the insertion of several Arg residues into the binding groove formed at the coiled-coil interface near the membrane-binding loop (*Figure 1—figure supplement 1*; *Cleverley et al., 2019*).

Our group previously found that *Sa* GpsB interacts with *Sa* FtsZ to regulate its polymerization characteristics, but the repeated (S/T/N)-R-X-X-R-(R/K) GpsB recognition motif on its C-terminus was not recognized until the work on such motifs in PBPs published by *Cleverley et al., 2019*. This motif is unique to *Staphylococci* FtsZ and is absent/less conserved in other Firmicutes (*Figure 3A*). It is unclear why the GpsB recognition motif is repeated, since there is only enough area in the PBP-binding groove to accommodate a helix of eight to nine residues. It is possible that *Sa* FtsZ binds two *Sa* GpsB dimers simultaneously, given the putative arrangement of GpsB as trimer of dimers (*Cleverley et al., 2016*). Alternatively, the first site may be occluded by the binding of other FtsZ interaction partners that are known to bind FtsZ through the C-term such as FtsA, EzrA, and SepF (*Huang et al., 2013*). This finding, in addition to our previous reports (*Eswara et al., 2018*; *Hammond et al., 2022*), underscore the importance of GpsB in *S. aureus* cell division.

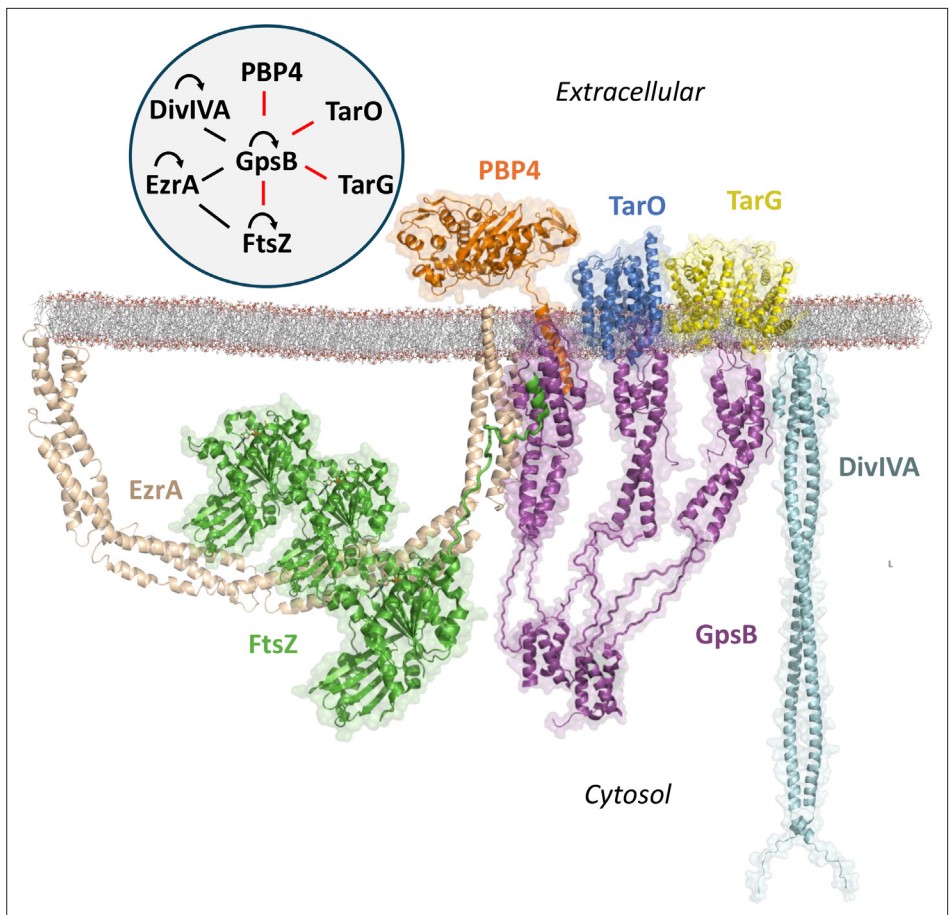

**Figure 5.** Known interactome of *Sa* GpsB and putative arrangement at the division septum in graphical and diagram (upper left) format. In this paper, we demonstrate that the C-terminal mini-domain of PBP4 (orange) and the C-terminus of FtsZ (green) bind to the N-terminal domain of GpsB (purple). The regions within GpsB responsible for interacting with other partners remain to be elucidated. For interaction diagram, the red lines indicate interactions putatively unique to *S. aureus*, black lines indicate interactions found in both *S. aureus* and *B. subtilis*, and curved arrows represent self-interaction.

A crystal structure of the *Sa* PBP4 C-terminal recognition sequence bound to the N-terminal domain of GpsB reveals a binding mode that mimics previously solved orthologous PBP/GpsB pairs. This discovery is notable because it was previously thought that only the N-terminal 'mini-domain' of transpeptidase PBPs (class A, class B) could bind to GpsB. Furthermore, this finding is significant not only because it is the sole *S. aureus* PBP that binds to GpsB, but also because *Sa* PBP4 is intimately associated with WTA synthesis (*Atilano et al., 2010*; *Farha et al., 2013*). Previous studies have found that *Sa* GpsB interacts with the WTA biosynthesis pathway proteins TarO (*Kent, 2024*) and TarG (*Hammond et al., 2022*), and likely facilitates the export of WTA to the septal cell wall. To our knowledge, *Sa* PBP4 does not directly interact with any of the known WTA machinery. However, it does require WTA for recruitment to the division septum, and impairment of WTA assembly results in delocalization of *Sa* PBP4 (*Atilano et al., 2010*). Thus, while *Sa* PBP4 itself is not essential for growth, it is tightly regulated by WTA synthesis, an essential process that is also mediated by GpsB (*Kent, 2024*; *Hammond et al., 2022*). Thus, it is conceivable that GpsB likely arrives at the division site together with FtsZ, facilitates WTA synthesis, and subsequently recruits PBP4 to promote efficient cytokinesis.

PBP4 joins several other known proteins found to interact with *Sa* GpsB: EzrA (*Steele et al., 2011*), DivIVA (*Bottomley et al., 2017*), FtsZ (*Eswara et al., 2018*), TarO (*Kent, 2024*), and TarG (*Hammond et al., 2022*; *Figure 5*). The interaction diagram outlines the known interactions for GpsB to date but given the diverse number of proteins at the divisome and various binding surfaces on GpsB, it will surely be expanded in the future. Furthermore, while GpsB lacks certain direct interactions with other

known divisome proteins, they are indirectly linked through intermediate proteins. For example, *Sa* FtsZ binds to *Sa* EzrA, a protein known to interact with the SEDS-PBP pairs *Sa* PBP1-FtsW and *Sa* PBP3-RodA, thus indirectly coupling *Sa* GpsB to enzymes that are critical for peptidoglycan synthesis at the divisome (**Steele et al., 2011**; **Reichmann et al., 2019**).

When evaluating the biophysical affinity of *Sa* FtsZ and *Sa* PBP4 for *Sa* GpsB, it is important to consider the peptide dissociation constants determined by SPR (~20–80 µM) may not directly translate to, and likely underestimate, the cellular affinity. The divisome is a highly complex environment with multiple proteins that interact near or at the cell membrane. The enrichment of both *Sa* FtsZ and *Sa* PBP4 at the division septum likely increases their apparent affinity simply based on avidity. Additionally, the arrangement of proteins may introduce synergistic interactions. Studies have found that EzrA, which binds to *Sa* GpsB (**Steele et al., 2011**), also binds to the C-terminus of FtsZ (**Buske and Levin, 2012**; **Singh et al., 2007**), likely upstream of the *Sa* GpsB recognition motif in *S. aureus*. This interaction could both increase the local concentration of *Sa* FtsZ and induce molecular recognition features that promote association with *Sa* GpsB. In addition, the oligomeric state of FtsZ and GpsB may further lead to cooperativity in the interactions between the FtsZ filament and GpsB hexamers. However, tighter binding may also prove deleterious in certain scenarios, especially for dynamic proteins like FtsZ. Given that *Sa* GpsB enhances the GTPase activity (required for FtsZ filament disassembly) of *Sa* FtsZ (**Eswara et al., 2018**), it is possible GpsB could dynamically promote polymerization and depolymerization of FtsZ.

Under the specter of growing antibacterial resistance, the identification of novel antibiotic targets is becoming increasingly urgent. Beyond PBPs, the bacterial divisome is a largely untapped source of antibiotic targets. The delineation of the physiological role of *Sa* GpsB, identification of its recognition motifs, and characterization of its 3D structure greatly enables modern antibiotic drug-discovery strategies. Furthermore, the involvement of *Sa* GpsB in multiple essential processes presents the opportunity to design targeted antibiotics. There is a growing need for narrow-spectrum antibiotics, especially for common infections, such as those caused by *S. aureus* (**Melander et al., 2018**). Narrow-spectrum antibiotics avoid selective pressure of commensal bacteria which can serve as a reservoir for resistance elements. In the same vein, their limited disruptive properties can avoid pathologies associated with bacterial dysbiosis, like *Clostridioides difficile* infection. The results from this study provide new information for both understanding the role of GpsB in *S. aureus* division and probing new avenues for narrow-spectrum antibiotic development.

## Methods

### Recombinant protein cloning and purification

The nucleotide sequence of *Sa gpsB* corresponding to the N-terminal domain (1–70) and full length (1–114) was inserted into a modified pET28a vector with a His-TEV sequence 5′ to the multiple cloning site (MCS). GpsB constructs expressed for SPR bioanalysis were cloned into a separate pET28a vector with a His-TEV-Avi sequence flanking the MCS (pAViBir). ΔMAD and ΔMNN mutants were generated with QuikChange site-directed mutagenesis using custom primers (ΔMAD (5′-GATTATCAAAAA ATGAATAATGAAGTTGTAAAATTATCAGAAGAGAATC) and ΔMNN (5′-GATTATCAAAAAATGGCCGA TGAAGTTGTAAAATTATCAGAAGAG)). All plasmids were transformed into Rosetta (DE3) pLysS cells. A single colony was grown in LB media supplemented with 35 µg/mL chloramphenicol and 50 µg/ mL kanamycin at 37°C overnight. The overnight culture was then diluted into 1 L LB media at 1:500 and incubated at 37°C until the $OD_{600}$ reached 0.8. Protein expression was initiated with 0.5 mM IPTG and continued incubation at 25°C overnight. pAViBir constructs were biotinylated during IPTG induction with a stock of 5 mM biotin dissolved in bicine for a final concentration of 50 µM. Cells were harvested by centrifugation at 5000 × *g* for 10 min. The cell pellet was resuspended in buffer A (20 mM Tris-HCl pH 8.0, 300 mM NaCl, 20 mM imidazole, and 10% glycerol). Cells were disrupted by sonication followed by centrifugation at 35,000 × *g* for 40 min. The pellet containing the protein was resuspended in buffer AD (100 mM Tris-HCl pH 8.0, 6 M guanidine HCl, 300 mM NaCl) and incubated at 30°C for ~1 hr to fully dissolve the pellet, followed by centrifugation at 45,000 × *g* for 1 hr. The supernatant was then loaded onto a HisTrap affinity column and eluted in a single step using Buffer BD (100 mM sodium acetate pH 4.5, 6 M guanidine HCl, 300 mM NaCl). The eluted protein was diluted dropwise in refolding buffer (100 mM Tris pH 8.0, 200 mM NaCl) and allowed to refold

overnight at 4°C. The sample was then loaded to a HisTrap column and eluted with linear gradient of imidazole. The fractions containing GpsB were pooled and concentrated. The protein was then incubated with TEV at 1:20 ratio overnight at 4°C. The sample was then loaded onto a HisTrap for reverse Ni$^{2+}$ cleanup. Flow-through was collected and purified using a HiLoad 16/60 Superdex 75 size exclusion column. The protein was stored at –80°C in the storage buffer (20 mM Tris-HCl pH 8.0, 200 mM NaCl). The purity of the protein was determined by SDS-PAGE as >95%. $^{15}$N-labeled GpsB (1-70) for NMR spectroscopy was expressed in the same way, but in minimal media containing $^{15}$N NH$_4$Cl and $^{12}$C glucose as the source of nitrogen and carbon, respectively, supplemented with MgSO$_4$, CaCl$_2$, and trace metals.

## Strain construction

Plasmids were generated using standard cloning procedures. The C-terminal six-residue truncation of FtsZ (FtsZ $^{\Delta C6}$) was generated by PCR using primer pairs oDB9/oDB10 (NdeI/XhoI) and cloned into the pET28a to create pDB1. FtsZ CTT encoding the C-terminal 66 amino acids of *S. aureus* FtsZ was cloned into pET28a using oP228/oP229 (Nde1/BamHI) to create pSK4. These were then transformed into BL21-DE3 cells creating EDB01 and SK7, respectively. ΔMAD was created by using site-directed mutagenesis with custom primers (5′-GATTATCAAAAAATGAATAATGAAGTTGTAAAATTATCAG AAGAGAATC) and ΔMNN was ordered from Integrated DNA Technologies as oLHgblock1. These mutations were then PCR-amplified and cloned into pDR111 to create both untagged (oP36/oP38; Hind111/Sph1) and GFP tagged (oP36/oP37; Hind111/Nhe1 and oP46/oP24; Nhe1/Sph1) variants. Plasmids were then transformed into PY79 and screened for amyE integration resulting in strains LH115, LH116, LH119, and LH126. The PCR products containing the ΔMAD and ΔMNN mutations were also cloned into pCL15 backbone using the same primers and restriction sites to create plasmids pLH62, pLH59, pLH63, and pLH60. These plasmids were then transformed into RN4220 cells resulting in strains LH129, LH127, LH130, and LH128. Then these plasmids were transduced into SH1000 to create strains LH135, LH134, LH133, and LH132. To make the BTH plasmids, ΔMAD and ΔMNN were amplified (BTH11/BTH12; EcoRI/XhoI) and cloned into pEB354 and pEB355 resulting in strains LH164, MA1, LH170, and LH168. The genotypes of strains and oligonucleotides used in this study are provided in ***Supplementary files 3 and 4***.

## CRISPR/Cas9-based gpsB deletion

Plasmid pCAS$^{SA}$ was used to delete *gpsB* from *S. aureus* RN4220 genome (***Chen et al., 2017***). Guide RNA (gRNA) targeting *gpsB* was designed with primer pairs oLM25/26. Primers were phosphorylated via T4 polynucleotide kinase treatment. Following phosphorylation, primers were annealed in the same reaction mixture supplemented with 2.5 µL of 1 M NaCl at 95°C for 3 min. Annealed gRNA was inserted upstream of *cas9* gene via compatible BsaI restriction sites utilized for Golden Gate Assembly. Upstream and downstream regions of *gpsB* were amplified for use in Gibson Assembly. The upstream region was amplified with primer pairs oLM27/28, containing restriction site XbaI compatible with pCAS$^{SA}$ and complementary to downstream sequence. The downstream region was ordered in a geneblock (gLM1 with internal XhoI site corrected) and amplified with primer pairs oLM29/30, containing complementarity to upstream (***Hammond et al., 2022***) sequence and restriction site XhoI compatible with pCAS$^{SA}$. pCAS$^{SA}$ containing gRNA was digested with XbaI and XhoI for linearization. Gibson Assembly was performed to insert amplified regions downstream of the gRNA scaffold. The resulting plasmid pLM29 was transformed into *Escherichia coli* DH5a. Correct conformation and insertion of sequences was confirmed via Sanger sequencing with primer pair oLM31/32. pLM29 was transformed into RN4220 via electroporation and incubated at 30°C for 24 hr. Colonies were screened with colony PCR for the amplification of *gpsB*. If *gpsB* was not detected, colony PCR was performed and the PCR product amplified using up- and downstream primers of gpsB locus was sent for Sanger sequencing to confirm *gpsB* deletion. Knockout strains were plasmid-cured by incubation of strains at 30°C overnight in 3 mL TSB. Overnight cultures were then diluted 1:1000 in TSB and incubated at 42°C until culture grew turbid. Confirmation of successful plasmid curing was determined by verifying susceptibility to 10 µg/mL chloramphenicol. Western blot analysis was also performed on knockout strains using α-GpsB antibody to ensure lack of GpsB production.

## X-ray crystallography

Crystals of GpsB (1–70) were grown in a hanging drop apparatus by mixing 11 mg/mL GpsB (purity >98%) with crystallization buffer (30% PEG 3350, 0.4 M NaCl, and 0.1 M Tris pH 8.5) in an equal ratio at 20°C. Filamentous crystals appeared overnight and were harvested after 1 week of growth by briefly transferring to cryoprotectant (30% PEG 3350, 0.4 M NaCl, 0.1 M Tris pH 8.5, and 15% glycerol), followed by flash freezing in liquid nitrogen. GpsB (1–70) and PBP4 (424–451) were mixed in a 1:1 ratio with a GpsB concentration of 6 mg/mL and 1.44 mM PBP4 peptide (1:1 ratio). Initial crystals grew in 25% PEG 4000, 0.1 M Tris pH 8.0, and 0.2 M sodium acetate. These crystals were crushed, diluted 10,000-fold, then seeded into drops in a ratio of 1:1:0.5 (protein, crystallization solution, seed stock). Crystals were harvested after 1 week of growth by transferring to a cryoprotectant solution of 27.5% PEG 4000, 0.1 M Tris pH 8.0, 0.2 M sodium acetate, and 15% glycerol. X-ray diffraction data were collected on the Structural Biology Center (SBC) 19-ID beamline at the Advanced Photon Source (APS) in Argonne, IL, and processed and scaled with the CCP4 versions of iMosflm (*Battye et al., 2011*) and Aimless (*Evans and Murshudov, 2013*). Initial models were obtained using the MoRDa (*Vagin and Lebedev, 2024*) package of the online CCP4 suite. Unmodeled regions were manually built and refined with Coot (*Emsley and Cowtan, 2004*).

## Circular dichroism

Thermal stability was assessed with CD using a Jasco J-815 CD spectropolarimeter coupled to a Peltier cell holder. Recombinantly expressed and purified WT GpsB and GpsB mutants were diluted to 2 µg/mL in 50 mM sodium phosphate (pH 7.0) and CD spectra were measured at 222 nm from 20°C to 80°C. Melting temperature was determined with a 4-parameter logistic curve fit using GraphPad Prism 9.

## Surface plasmon resonance

A Series S CM5 chip (Cytiva) was docked into a Biacore S200 instrument (Cytiva) followed by surface activation with NHS/EDC amine coupling. Lyophilized neutravidin (Thermo Fisher Scientific) was dissolved in sodium acetate pH 5.25 to a final concentration of 0.25 mg/mL and injected onto the activated CM5 chip at 10 µL/min for 5 min. Biotinylated GpsB was diluted to 1 mg/mL in HEPES-buffered saline (HBS) and injected over the neutravidin-immobilized CM5 chip at 20 µL/min for 5 min. Synthetic peptides and recombinantly expressed *Sa* FtsZ (325–390) were serially diluted in HBS, and injected at 30 µL/min for 50 s with a dissociation of 100 s, followed by a stabilization period of 15 s and a buffer wash between injections. All experiments were performed in technical duplicate. Dissociation constants were determined with a one site binding model using GraphPad Prism 9.

## NMR spectroscopy

2D $^1$H-$^{15}$N HSQC spectra of *Sa* GpsB (1–70) were recorded on an Agilent 800-MHz direct drive instrument equipped with a cryoprobe. NMRpipe (*Delaglio et al., 1995*) and Sparky (University of California, San Francisco) were used for processing and analysis, respectively. All spectra were acquired in 20 mM Tris pH 8.0, 200 mM NaCl, prepared in 7.5% $D_2O$, and at 25°C. The concentration of *Sa* GpsB$^{WT}_{1-70}$ was 160 or 700 µM for the free protein spectrum and 160 µM for the complex with the *Sa* FtsZ octapeptide, which was added in a 1.5× molar excess.

## GTP hydrolysis of FtsZ

*Sa* FtsZ, *Sa* GpsB, and *Sa* FtsZ$^{ΔC6}$ were purified using Ni-NTA affinity chromatography as described previously (*Eswara et al., 2018*). The effect of GpsB on the GTPase activity of FtsZ and FtsZ$^{ΔC6}$ was determined by measuring the free phosphate released using the malachite green phosphate assay kit (Sigma-Aldrich, MAK307-1KT). Briefly, either FtsZ or FtsZ$^{ΔC6}$ (30 µM) was incubated with GpsB (10 µM) in the polymerization buffer (20 mM HEPES pH 7.5, 140 mM KCl, 5 mM $MgCl_2$) containing 2 mM GTP at 37°C for 15 min. The free phosphate released was determined by measuring the absorbance of the reaction mixture at 620 nm. Statistical analysis was completed using GraphPad Prism 9 with the one-way ANOVA and post hoc multiple comparisons.

### *B. subtilis* and *S. aureus* growth conditions

Liquid cultures of *B. subtilis* cells were grown in LB and *S. aureus* cells were grown in TSB supplemented with 10 µg/mL chloramphenicol at 37°C.

### Spot titer assays

Overnight cultures of *B. subtilis* and *S. aureus* strains were back diluted to $OD_{600}$=0.1 and grown to midlog phase ($OD_{600}$=0.4). Cultures were then back diluted to an $OD_{600}$ of 0.1, serial diluted, and spotted onto LB plates with or without 1 mM IPTG (*B. subtilis*), or TSA plates supplemented with 10 µg/mL chloramphenicol with or without 1 mM IPTG (*S. aureus*), and incubated overnight at 37°C. Spot titers were repeated twice in technical triplicate each time.

### Fluorescence microscopy

Overnight cultures of *B. subtilis* and *S. aureus* strains to be imaged were back diluted to $OD_{600}$=0.1 and grown to midlog phase ($OD_{600}$=0.4) and then induced with 1 mM IPTG and allowed to grow for an additional 2 hr. Cells were then prepared and imaged as previously described (*Brzozowski et al., 2019*). Briefly, 1 mL aliquots were spun down, washed, and resuspended in PBS. Cells were then stained with 1 µg/mL SynaptoRed fluorescent dye (MilliporeSigma, 574799-5MG) to visualize the membrane and 5 µL of culture was spotted onto a glass bottom dish (Mattek, P35G-1.5-14-C). Images were captured on a DeltaVision Core microscope system (Leica Microsystems) equipped with a Photometrics CoolSnap HQ2 camera and an environmental chamber. Seventeen planes were acquired every 200 nm and the data were deconvolved using SoftWorx software. Cells were measured using ImageJ and analyzed using GraphPad Prism 9 with the one-way ANOVA and post hoc multiple comparisons.

### Bacterial two-hybrid assay

Plasmids carrying genes of interest cloned into the pEB354 (T18 subunit) and pEB355 (T25 subunit) backbones were transformed pairwise into BTH101 cells. Overnight cultures of the strains grown in 100 µg/mL ampicillin, 50 µg/mL kanamycin, and 0.5 mM IPTG, at 30°C, were spotted onto MacConkey agar containing 1% maltose that were also supplemented with ampicillin, kanamycin, and IPTG. Plates were incubated for 24 hr at 30°C and then imaged. Two biological replicates were performed each in technical triplicate. The β-galactosidase assay was carried out as previously described (*Hammond et al., 2022*). Mixtures of 20 µL of culture, 30 µL of LB, 150 µL Z buffer, 40 µL ONPG (4 mg/mL), 1.9 µL β-mercaptoethanol, and 95 µL polymyxin B (20 mg/mL) were transferred to a 96-well plate and read on a BioTek plate reader. Miller units were then calculated and graphed using GraphPad Prism 9 with the one-way ANOVA and post hoc multiple comparisons.

### Immunoblot

Overnight cultures of *S. aureus* cells were back diluted to $OD_{600}$=0.1, grown to midlog phase ($OD_{600}$=0.4), and then induced with 1 mM IPTG and grown for an additional 2 hr. Cells were then standardized to an $OD_{600}$=1.0, lysed with 5 µL lysostaphin (1 mg/mL in 20 mM sodium acetate), and incubated for 30 min at 37°C. 1 µL of DNAse A (1 U/µL) was added and incubated for an additional 30 min. Samples were then analyzed by SDS-PAGE analysis, transferred to a membrane, and probed with rabbit antiserum raised against GpsB-GFP. Total protein was visualized from the SDS-PAGE gel using the GelCode Blue Safe Protein Stain (Thermo Fisher, 24596).

### Homology model construction of deletion variants

Template-based homology models were made using MODELLER 9.24 (*Eswar et al., 2006*) by constructing 1000 decoys corresponding to each construct based on various template structures.

### MD simulations

The GpsB dimers were exposed to conventional MD simulation using Gromacs (v. 5.0.4) (*Abraham et al., 2015*; *Páll et al., 2014*) with the CHARMM36m force field (*Huang et al., 2017*). Explicit TIP3P water (*Jorgensen et al., 1983*) with 150 mM KCl was used for solvation. A 12 Å cut-off for the van der Waals forces was used. Electrostatic forces were computed using the particle mesh Ewald method (*Darden et al., 1993*). The Verlet cut-off scheme was used. The temperature and pressure were controlled using the Nosé-Hoover (*Nosé, 1984b*; *Nosé, 1984a*; *Hoover, 1985*; *Parrinello*

*and Rahman, 1981*; *Nosé and Klein, 1983*) methods, respectively, to sample the NPT ensemble at P=1 bar and T=303.15 K. The integration time step was 2 fs, enabled by using H-bond restraints (*Hess et al., 1977*). Each system was simulated for 250–500 ns. All systems were made using CHARMM-GUI (*Jo et al., 2008*; *Brooks et al., 2009*; *Lee et al., 2016*).

## AlphaFold2 predictions

Prediction of the complex between the GpsB dimer and the C-terminal domain of FtsZ was performed using AlphaFold2 (multimer version 3) (*Evans et al., 2021*; *Varadi et al., 2022*) on ColabFold (*Mirdita et al., 2022*; *Steinegger and Söding, 2017*). The sequence of GpsB was taken from *S. aureus* (strain COL, UniProt ID: Q5HFX8) and for FtsZ we used *S. aureus* (strain COL, UniProt ID: Q5HGP5) residues 325–390 for the entire C-terminal domain or residues 385–390 for the terminal 6-peptide. Structural predictions were made using all standard options on ColabFold and seeding the template structure resolved in this study between GpsB and the PBP4 fragment (PDB ID: 8E2C) was not observed to improve predictions.

## Acknowledgements

We thank Eric Lewandowski for reading the manuscript. We also thank the staff members of the Advanced Photon Source of Argonne National Laboratory, particularly those at the SBC for assistance with X-ray diffraction data collection. SBC-CAT is operated by UChicago Argonne LLC, for the U.S. Department of Energy, Office of Biological and Environmental Research under contract DE-AC02-06CH11357.

## Additional information

### Funding

| Funder | Grant reference number | Author |
| --- | --- | --- |
| National Institutes of Health | R21 AI164775 | Yu Chen Prahathees Eswara |
| National Institutes of Health | R35 GM133617 | Prahathees Eswara |

The funders had no role in study design, data collection and interpretation, or the decision to submit the work for publication.

### Author contributions

Michael D Sacco, Conceptualization, Data curation, Formal analysis, Investigation, Visualization, Methodology, Writing - original draft, Writing - review and editing; Lauren R Hammond, Conceptualization, Data curation, Formal analysis, Investigation, Writing - original draft, Writing - review and editing; Radwan E Noor, Dipanwita Bhattacharya, Lily J McKnight, Jesper J Madsen, Xiujun Zhang, Shane G Butler, M Trent Kemp, Aiden C Jaskolka-Brown, Sebastian J Khan, Investigation; Ioannis Gelis, Formal analysis, Investigation, Writing - original draft; Prahathees Eswara, Yu Chen, Conceptualization, Resources, Formal analysis, Supervision, Funding acquisition, Investigation, Methodology, Writing - original draft, Project administration, Writing - review and editing

### Author ORCIDs

Michael D Sacco http://orcid.org/0000-0001-5930-7363
Dipanwita Bhattacharya http://orcid.org/0000-0002-5000-3846
Jesper J Madsen http://orcid.org/0000-0003-1411-9080
Prahathees Eswara http://orcid.org/0000-0003-4430-261X
Yu Chen http://orcid.org/0000-0002-5115-3600

### Decision letter and Author response

Decision letter https://doi.org/10.7554/eLife.85579.sa1
Author response https://doi.org/10.7554/eLife.85579.sa2

## Additional files

### Supplementary files
• Supplementary file 1. Table of crystallographic statistics. *Values in parentheses indicate those for the highest resolution shell.

• Supplementary file 2. Surface plasmon resonance (SPR) dissociation constants ($K_D$) of *Sa* FtsZ and *Sa* PBP4 derived peptides for *Sa* GpsB$^{WT}_{FL}$ and *Sa* GpsB$^{\Delta MAD}_{FL}$. Concentration response sensorgrams are shown in *Figure 4—figure supplement 2*.

• Supplementary file 3. The genotypes of strains used in the cell-based studies.

• Supplementary file 4. The oligonucleotide and geneblock sequences used in the cell-based studies.

• MDAR checklist

### Data availability
All crystal structures have been deposited in the RCSB Protein Data Bank (PDB) with accession IDs of: Sa GpsB NTD (PDB ID 8E2B), Sa GpsB NTD + Sa PBP4 C-term (PDB ID 8E2C).

The following datasets were generated:

| Author(s) | Year | Dataset title | Dataset URL | Database and Identifier |
| --- | --- | --- | --- | --- |
| Sacco M, Chen Y | 2023 | N-terminal domain of *S. aureus* GpsB | https://www.rcsb.org/structure/8E2B | RCSB Protein Data Bank, 8E2B |
| Sacco M, Chen Y | 2023 | N-terminal domain of *S. aureus* GpsB in complex with PBP4 fragment | https://www.rcsb.org/structure/8E2C | RCSB Protein Data Bank, 8E2C |

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
