## [Editor Report]

This valuable work reports a unique N-terminal motif of *Staphylococcus aureus* GpsB, the co-crystal structure of GpsB with the C-terminus of PBP4, and the direct interaction between GpsB and the C-terminus of FtsZ. The evidence supporting these discoveries is convincing, with biochemical and structural characterizations. This study sheds light on the role of GpsB in the cell division of this important pathogen.

---

## [Decision Letter]

**Decision letter after peer review:**

Thank you for submitting your article "*Staphylococcus aureus* FtsZ and PBP4 bind to the conformationally dynamic N-terminal domain of GpsB" for consideration by *eLife*. Your article has been reviewed by 3 peer reviewers, one of whom is a member of our Board of Reviewing Editors, and the evaluation has been overseen by Bavesh Kana as the Senior Editor. The following individual involved in the review of your submission has agreed to reveal their identity: Tobias Doerr (Reviewer #3).

Essential revisions (for the authors):

The reviewers identified that the study of S. aurues GpsB and its interactions with FtsZ and PBP4 is important for understanding the essential role of GpsB in the cell division of *S. aureus*. However, the structure of GpsB and PBPs has been reported before in *B. subtilis* and *S. pneumoniae*, and the functional characterization of GpsB's new motif and the structural characterization of GpsB and FtsZ's interaction is incomplete. The manuscript requires extensive revisions to address two essential points raised by the reviewers.

1) Please address the functionality of the 3-aa insertion/kinked structure motif in *S. aureus* GpsB by providing new experiments as requested by Reviewer #1, points 1 and 2, and Reviewer #3, point 1 (first paragraph of recommendations to authors).

2) Please provide a structural characterization of GpsB in complex with the C-terminus of FtsZ. If a co-structure is not possible, please provide justifications and/or a discussion on a predicated model using Alphafold2/RosettaFold.

*Reviewer #1 (Recommendations for the authors):*

1. The crystal structure revealed similarities and differences between Sa GpsB and GpsB from other species. Sa GpsB contained a hinge in the helix that results in bending of the protomer. This feature was not observed in other species. The authors provided strong evidence showing that the hinge was not a crystallization artifact but a novel motif through stability measurements, NMR, and MD simulations (Figures 1 and 2). The authors further tested the effects of deletion of two 3-residue insertions uniquely present in Sa GpsB (∆MAD and ∆MNN) using the lethality of Sa GpsB overexpression in *B. subtilis* (Figure 2). While these experiments are important, the overexpression of ∆MAD or ∆MNN in *S. aureus* appeared to have a modest effect (Figure S4), thus it may be difficult to argue that the hinge is important for Sa GpsB function. The authors hypothesized that this modest effect is due to the differential affinity of ∆MNN to itself than Sa GpsB, but it is unclear how it would explain the modest effect. Furthermore, these mutational investigations may be better/easier to interpret if mutants are expressed in wt gpsB depletion background.

2. To complement these experiments, perhaps the authors could insert MAD or MNN into Bs GpsB and monitor the mutants' thermostability, localization, and lethality in *B. subtilis* subsequently. Additionally, overexpressed WT Sa-GpsB mislocalized in *B. subtilis* and was lethal, while overexpressed ∆MAD and ∆MNN localized to the mid cell and cells were normal. Do these results indicate that the lethality of SaGpsB in *B. subtilis* is a nonspecific protein aggregation effect and that the two regions are responsible for the aggregation? What protein-protein interactions contribute to the mid-cell localization of SaGpsB in *B. subtilis*? If the toxicity of Sa GpsB in Bs is caused by the hinge structure one would expect mutating the Bs GpsB to include it would also have a similar toxic effect.

3. The discovery that the CTV of Sa FtsZ has a repeat match of the consensus GpsB binding motif is interesting. The binding affinities at ~ 20 to 70 μM, however, do not appear to be strong. The authors suggested that cellular affinities may be higher and a regulatory point. Could the authors provide a negative control using Sa FtsZ∆6, or a peptide from another species that does not have the motif to rule out sequence-independent binding? These controls may be important because the FtsZ (320-390) terminus includes the two negatively charged E but binds tighter than FtsZ (379-390), which also contains the two E residues. Additionally, if we assume that the crystallographic work between the FtsZ terminus with Sa GpasB is not successful, can a structure be deduced from the NMR study?

4. As the FtsZ CTV's binding motif is similar to that of PBP4, do they compete with each other, or can they both bind to two monomers of GpsB simultaneously? How important is this binding? The authors reported that ∆MAD has lost the binding to PBP4 and FtsZ CTV, but it is hard to imagine how the structure that is different from the binding site causes a significant reduction in binding. To demonstrate the importance of binding, the authors may wish to design some mutations at the binding surface and exam the consequences in cell physiology.

*Reviewer #2 (Recommendations for the authors):*

While I appreciate the efforts in providing a solid base for the characterization of the interactions between GspB and FtsZ or PBP4; I consider the novelty is not enough to publish in *eLife* considering the previously published works.

– Results. When describing the 3D structure of the GspB N-term domain; Did authors run AlphaFold2 (AF2) to see the prediction of the hexameric full-length structure? Is this configuration compatible with interaction with other partners? How the discovered hinge could affect the oligomeric arrangement of the full-length protein? How these results could compare with GspB from other bacterial species for which this region has been also solved?

– Results. Line 201. Also, as there is no 3D structure for the complex between the N-term domain of GspB and the C-term domain of FtsZ, AF2 prediction could be important to identify if the same pattern of interactions observed for PBP4 are observed here, and maybe to identify key residues in this interaction. Mutagenesis experiments could then validate this interaction.

– Results. When describing the interactions between GspB and PBP4, authors should directly compare with previous interactions observed for GspB in *B. subtilis* and *S. pneumoniae* (Cleverley et al. Nat Comms 2019). Now, this information is only partially presented when comparing crystal packing in *S. aureus* GspB with the complex GspB:PBP1a in *B. subtilis*.

Two Arg residues seem to be critical in the interaction with PBPs, is this interaction lost if you mutated both of them?

– Figure 5. Please indicate how this model was generated. Is this just an artistic representation of the partners? is based on previous structures? or on predictions by AF2?

*Reviewer #3 (Recommendations for the authors):*

This is a nice study overall, and very well-written and well-organized. My main issue is with the overexpression experiments. Overexpression toxicity of divisome components can be highly pleiotropic and is difficult to interpret. I don't follow the conclusion that the insertion sequence is important for function since an assessment of functionality is only based on overexpression toxicity. Can you replace native GpsB with the ∆MAD and ∆MNN mutants? This would be the ultimate test of functionality. If not possible, you could conduct depletion experiments of the wild-type copy in a background expressing the mutants.

It is unclear to me why the ∆MAD/MNN mutants are less toxic than the WT (see my minor comment below as well). I don't find the bacterial two-hybrid data very convincing. BACTH is not necessarily quantitative, so more precise experiments (ELISA?) would be necessary to conclude something about the affinity of heterocomplexes. Since the exact mechanism of toxicity is not important for their conclusions, maybe tread a little more lightly here. The statement in line 185 is a bit too strong.

Figure 3C needs a negative control. Not knowing if a value of ~1.5 in their GTPase assay is significantly above the background, it is possible that the ∆C6 mutant is simply catalytically dead, and thus cannot be activated by GpsB. A GTPase point mutant in FtsZ would be a great control here to establish whether FtsZ without GpsB has activity significantly above background, and would consequently demonstrate that the ∆C6 mutant is specifically deficient in GpsB-mediated activation.

[Editors' note: further revisions were suggested prior to acceptance, as described below.]

Thank you for resubmitting your work entitled "Staphylococcus aureus FtsZ and PBP4 bind to the conformationally dynamic N-terminal domain of GpsB" for further consideration by *eLife*. Your revised article has been evaluated by Bavesh Kana (Senior Editor) and a Reviewing Editor.

The manuscript has been improved but there are some remaining issues that need to be addressed, as outlined below:

Please add discussions regarding the putative arrangement between FtsZ and GpsB in Figure 5, either by Alpah-fold2 predication as Rev #2 suggested, or textual justifications.

*Reviewer #2 (Recommendations for the authors):*

The authors have partially responded to my previous review.

It seems that more detailed information on the C-term FstZ and GpsB interaction is ongoing for future work.

The predicted model by AF multimer of the N-term GpsB dimer and the C-term of FtsZ is straightforward to do, and this model could reinforce, at this moment, the experimental results provided in the manuscript. If the AF multimer fails, thus it can be just mentioned in the manuscript.

Other potential implications of this in silico experiment, could be to know if the GpsB dimer interacts with one or two FtsZ chains.

Also, as a proof of concept, the same AF multimer run can be done with PBP4 and GspB to see if a more extended picture of the PBP4-GspB complex can be reached.

*Reviewer #3 (Recommendations for the authors):*

The authors have adequately responded to the first round of reviews. I still believe that an FtsZ GTPase mutant would be the better control for the biochemical assay showing stimulation of GTPase activity by GpsB, but BSA is adequate.

I also still believe that a true examination of the functional importance of the hinge domain residues would require replacement of the native GpsB with these variants, followed by phenotypic characterization (though my understanding is that ∆gpsB phenotypes are subtile, but measurable nonetheless via e.g. morphology). The current reliance on overexpression toxicity (which was pointed out during the first round of reviews) makes interpretation of these data more difficult. That said, using a gpsB mutant for overexpression experiments is certainly an improvement over the first version of the manuscript.

---

## [Author Response]

Essential revisions (for the authors):Reviewer #1 (Recommendations for the authors):1. The crystal structure revealed similarities and differences between Sa GpsB and GpsB from other species. Sa GpsB contained a hinge in the helix that results in bending of the protomer. This feature was not observed in other species. The authors provided strong evidence showing that the hinge was not a crystallization artifact but a novel motif through stability measurements, NMR, and MD simulations (Figures 1 and 2). The authors further tested the effects of deletion of two 3-residue insertions uniquely present in Sa GpsB (∆MAD and ∆MNN) using the lethality of Sa GpsB overexpression in *B. subtilis* (Figure 2). While these experiments are important, the overexpression of ∆MAD or ∆MNN in *S. aureus* appeared to have a modest effect (Figure S4), thus it may be difficult to argue that the hinge is important for Sa GpsB function. The authors hypothesized that this modest effect is due to the differential affinity of ∆MNN to itself than Sa GpsB, but it is unclear how it would explain the modest effect. Furthermore, these mutational investigations may be better/easier to interpret if mutants are expressed in wt gpsB depletion background.

We agree the interpretation of the functionality of the hinge region mutants in the presence of native wt GpsB is not ideal. Inspired by the new reports, within the past few months, that showed *gpsB* can be deleted (PMID: 37711690 | bioRxiv: 538170; 545294), we generated our own *gpsB* knockout using CRISPR-Cas9 approach in *S. aureus* (PMID: 28218837). Using this del-*gpsB* strain, we now show that ∆MAD and ∆MNN variants of GpsB are partially and less functional respectively (Figure 2 Supplement 2A). Thus, we believe that the hinge region is important for *S. aureus* GpsB function by dynamically controlling GpsB activity and/or possibly enabling interactions with other partners.

Previously we showed that the toxic effect of GpsB is suppressed by co-expression of non-functional *gpsB* alleles in *B. subtilis* (Hammond et al., Microbiology Spectrum 2022; Figure S2A). As shown in Figure 2 Supplement 2D in this revised manuscript, co-expression of ∆MAD or ∆MNN does not fully inhibit the function of unmutated GpsB (although individually they are not toxic; Figure 2C). This implies that the flexibility provided by the hinge region allows for additional control of GpsB function within the GpsB complex (Figure 1D).

2. To complement these experiments, perhaps the authors could insert MAD or MNN into Bs GpsB and monitor the mutants' thermostability, localization, and lethality in *B. subtilis* subsequently. Additionally, overexpressed WT Sa-GpsB mislocalized in *B. subtilis* and was lethal, while overexpressed ∆MAD and ∆MNN localized to the mid cell and cells were normal. Do these results indicate that the lethality of SaGpsB in *B. subtilis* is a nonspecific protein aggregation effect and that the two regions are responsible for the aggregation? What protein-protein interactions contribute to the mid-cell localization of SaGpsB in *B. subtilis*? If the toxicity of Sa GpsB in Bs is caused by the hinge structure one would expect mutating the Bs GpsB to include it would also have a similar toxic effect.

Thank you for suggesting this idea. We previously conducted a suppressor screen to identify the source of toxicity in *B. subtilis* (Hammond et al., Microbiology Spectrum 2022). We reported that inhibition of wall teichoic acids pathway relieves the toxicity. We are also exploring possible interactions between Sa GpsB and Bs FtsZ in light of the new data from this manuscript. At lower inducer concentration, Sa GpsB localizes to mid-cell (Eswara et al., *eLife* 2018; Figure 1—figure supplement 2). Therefore, we do not think non-specific aggregation is the cause of toxicity. Multiple other factors are different between Bs GpsB and Sa GpsB besides MAD/MNN such as linker length and potential affinity to FtsZ. Thus, we did not test whether MAD/MNN insertion in Bs GpsB renders it toxic, as it is physiologically not relevant, and a negative result would not be informative due to the presence of other differences between Bs GpsB and Sa GpsB.

3. The discovery that the CTV of Sa FtsZ has a repeat match of the consensus GpsB binding motif is interesting. The binding affinities at ~ 20 to 70 μM, however, do not appear to be strong. The authors suggested that cellular affinities may be higher and a regulatory point. Could the authors provide a negative control using Sa FtsZ∆6, or a peptide from another species that does not have the motif to rule out sequence-independent binding? These controls may be important because the FtsZ (320-390) terminus includes the two negatively charged E but binds tighter than FtsZ (379-390), which also contains the two E residues. Additionally, if we assume that the crystallographic work between the FtsZ terminus with Sa GpasB is not successful, can a structure be deduced from the NMR study?

In Figure 4 and the main text, we discussed the testing of N-terminal minidomain peptides of Sa PBP1, PBP2, PBP2A (MRSA), and PBP3 –which lack this motif. These sequences are shown in Figure 4a. We found there was no dose dependent saturation, suggesting no sequence independent binding occurs. Furthermore, the GTPase assay (Figure 3C) comparing Sa FtsZ with Sa FtsZ∆6, supports the notion that these terminal six residues are critical for function. Although complex structure characterization using the current FtsZ peptides (residues 379-390, 383-390) has been unsuccessful, we have been actively testing crystallization using shorter peptides corresponding to each binding motif. Ultimately we feel a complex crystal structure will be possible and provide more information than NMR analysis. The data from the current NMR study is also not enough for deriving any detailed structural information.

4. As the FtsZ CTV's binding motif is similar to that of PBP4, do they compete with each other, or can they both bind to two monomers of GpsB simultaneously? How important is this binding? The authors reported that ∆MAD has lost the binding to PBP4 and FtsZ CTV, but it is hard to imagine how the structure that is different from the binding site causes a significant reduction in binding. To demonstrate the importance of binding, the authors may wish to design some mutations at the binding surface and exam the consequences in cell physiology.

As speculated in our model (Figure 5), we believe the GpsB complex would be able to bind FtsZ and PBP4 simultaneously, each by a different GpsB protomer and with similar K_D_ values determined by SPR. As shown by Supplementary File 2, ∆MAD affected GpsB binding by different FtsZ/PBP4 peptides slightly differently, ranging from ~2 to >4 fold. Due to the limitations of our SPR analysis, two K_D_ values were shown as >200 µM, but these values may reflect only a 3-4 fold decrease of the binding affinity. These data suggest that the ∆MAD retained binding of the partner peptides but with weaker affinities, probably due to unfavorable interactions occurring outside the binding pocket for the GpsB recognition motif as a result of the altered tertiary structure in the mutant. We agree that it will be informative to investigate how the interactions involving the GpsB recognition motif may affect GpsB function, as shown by the data concerning Sa FtsZ∆6. The overall importance of similar interactions was demonstrated by previous mutagenesis studies of GpsB from other species (PMID: 30651563). We intend to perform more comprehensive analysis of these interactions in *S. aureus* in the future.

Reviewer #2 (Recommendations for the authors):While I appreciate the efforts in providing a solid base for the characterization of the interactions between GspB and FtsZ or PBP4; I consider the novelty is not enough to publish in eLife considering the previously published works.

We wish to point out that the innovation and significance of our results goes beyond simply the structural basis of GpsB protein-protein interactions. The conformational heterogeneity of *S. aureus* GpsB is not observed in any of the other previously determined homolog structures, and this information will be important for our understanding of its unique role in *S. aureus* growth compared with the less critical functions of the homologs in other bacteria. The location of the GpsB-recognition motif in *S. aureus* proteins (i.e., C-terminus) is also different from all the previous examples (N-terminus), and our results suggest alternative binding patterns that will need to be analyzed in future studies. Finally, considering the importance of FtsZ in cell division, the identification of the specific FtsZ sequence bound by GpsB can lead to new experiments on how the Z-ring is formed and regulated in coordination with other cellular processes. Also, GpsB-FtsZ interaction has not been reported in other organisms.

– Results. When describing the 3D structure of the GspB N-term domain; Did authors run AlphaFold2 (AF2) to see the prediction of the hexameric full-length structure? Is this configuration compatible with interaction with other partners? How the discovered hinge could affect the oligomeric arrangement of the full-length protein? How these results could compare with GspB from other bacterial species for which this region has been also solved?

We previously used AlphaFold2 to predict the structure of the dimer of the GpsB N-terminal domain. However, the program failed to predict the kink we observed in our experimental structures. We do not think the hinge region will affect the oligomeric arrangement of the full-length protein because the two GpsB domains are connected by a flexible linker region and thus likely function independently of one another. Our experiments have shown that the dimer of the *S. aureus* GpsB N-terminal domain is able to interact with multiple proteins, so the hinge region does not appear to hinder GpsB protein-protein interactions.

– Results. Line 201. Also, as there is no 3D structure for the complex between the N-term domain of GspB and the C-term domain of FtsZ, AF2 prediction could be important to identify if the same pattern of interactions observed for PBP4 are observed here, and maybe to identify key residues in this interaction. Mutagenesis experiments could then validate this interaction.

We agree that it will be important to understand the molecular basis for the GpsB-FtsZ interactions. As described in the main text, the GpsB-binding sequence in FtsZ appears unique as it has two similar copies of the canonical recognition motif. We do plan to carry out an in-depth investigation, including mutagenesis, biochemical and structural studies, into a number of questions concerning FtsZ binding, such as how both recognition motifs may contribute to GpsB binding, whether the C-terminal carboxylate group plays a role in binding, and whether the C-terminus of *B. subtilis* FtsZ binds to GpsB as well. Due to the focus of the current manuscript and the amount of analysis to be performed, we feel these questions can be better addressed in a follow-up paper. However, we have added discussion in the current paper to briefly address the need for such future experiments. Also, as demonstrated by our preliminary analysis, AF2 could not capture important structural details unique to *S. aureus* GpsB. It is also not particularly suitable for protein-protein docking. Therefore we did not use AF2 to generate a model, although we feel that some important contacts, such as those involving the arginines, may be retained in FtsZ-GpsB interactions, based on the GpsB complex structures by us and others.

– Results. When describing the interactions between GspB and PBP4, authors should directly compare with previous interactions observed for GspB in *B. subtilis* and *S. pneumoniae* (Cleverley et al. Nat Comms 2019). Now, this information is only partially presented when comparing crystal packing in *S. aureus* GspB with the complex GspB:PBP1a in *B. subtilis*.Two Arg residues seem to be critical in the interaction with PBPs, is this interaction lost if you mutated both of them?

Both arginine residues form multiple interactions with GpsB in the superimposed structures of *S. aureus* and *B. subtilis* complexes in Figure 4B, although the first arginine is placed deeper into the binding pocket and may thus be more critical for GpsB binding. Based on the structural data and the conserved nature of these two residues in the GpsB-binding sequences, we agree that these arginine residues would make important contributions to GpsB binding and the interactions would be nearly abolished if both are mutated. The amount of new information generated by mutating these two residues will be relatively limited. But as we mentioned above, we plan to carry out a detailed mutagenesis analysis of the whole CTV region of FtsZ in the future, which will address these questions as well as those involving the two copies of GpsB-binding motif.

– Figure 5. Please indicate how this model was generated. Is this just an artistic representation of the partners? is based on previous structures? or on predictions by AF2?

This is an artistic representation of the interactions using previously determined structures of individual proteins. We have added clarifications in the figure legend.

Reviewer #3 (Recommendations for the authors):This is a nice study overall, and very well-written and well-organized. My main issue is with the overexpression experiments. Overexpression toxicity of divisome components can be highly pleiotropic and is difficult to interpret. I don't follow the conclusion that the insertion sequence is important for function since an assessment of functionality is only based on overexpression toxicity. Can you replace native GpsB with the ∆MAD and ∆MNN mutants? This would be the ultimate test of functionality. If not possible, you could conduct depletion experiments of the wild-type copy in a background expressing the mutants.

We wish to thank this reviewer for their kind words. Please see our response to comment #1 of reviewer #1. Briefly, we used a del-gpsB strain to study the functionality of GpsB hinge mutants. Our results show that ∆MAD and ∆MNN variants are partially and less functional respectively (Figure 2 Supplement 2A). We believe that the hinge region may result in different GpsB conformers with various activities to dynamically finetune GpsB function, and/or allow for interaction with other partners.

It is unclear to me why the ∆MAD/MNN mutants are less toxic than the WT (see my minor comment below as well). I don't find the bacterial two-hybrid data very convincing. BACTH is not necessarily quantitative, so more precise experiments (ELISA?) would be necessary to conclude something about the affinity of heterocomplexes. Since the exact mechanism of toxicity is not important for their conclusions, maybe tread a little more lightly here. The statement in line 185 is a bit too strong.

As shown in Figure 2 Supplement 2A, ∆MAD and ∆MNN are less toxic. We previously showed that the toxic effect of GpsB is suppressed by co-expression of non-functional *gpsB* alleles in *B. subtilis* (Hammond et al., Microbiology Spectrum 2022; Figure S2A). In Figure 2 Supplement 2D we show that co-expression of ∆MAD or ∆MNN does not fully inhibit the function of unmutated GpsB (although individually they are not toxic; Figure 2C). This implies that the flexibility provided by the hinge region may allow for additional control of GpsB function within the GpsB complex (Figure 1D). We have now moderated our interpretation of our BACTH results as per your suggestion.

Figure 3C needs a negative control. Not knowing if a value of ~1.5 in their GTPase assay is significantly above the background, it is possible that the ∆C6 mutant is simply catalytically dead, and thus cannot be activated by GpsB. A GTPase point mutant in FtsZ would be a great control here to establish whether FtsZ without GpsB has activity significantly above background, and would consequently demonstrate that the ∆C6 mutant is specifically deficient in GpsB-mediated activation.

As FtsZ and FtsZ∆C6 both show similar GTPase activity we do not think ∆C6 is catalytically dead. To alleviate this concern further, we repeated the experiments with Bovine Serum Albumin (BSA) as an additional control (data shown in Author response image 1). BSA did not have any GTPase activity, as expected.

**Author response image 1. sa2fig1:** 

[Editors’ note: what follows is the authors’ response to the second round of review.]

The manuscript has been improved but there are some remaining issues that need to be addressed, as outlined below:Please add discussions regarding the putative arrangement between FtsZ and GpsB in Figure 5, either by Alpah-fold2 predication as Rev #2 suggested, or textual justifications.Reviewer #2 (Recommendations for the authors):The authors have partially responded to my previous review.It seems that more detailed information on the C-term FstZ and GpsB interaction is ongoing for future work.The predicted model by AF multimer of the N-term GpsB dimer and the C-term of FtsZ is straightforward to do, and this model could reinforce, at this moment, the experimental results provided in the manuscript. If the AF multimer fails, thus it can be just mentioned in the manuscript.Other potential implications of this in silico experiment, could be to know if the GpsB dimer interacts with one or two FtsZ chains.Also, as a proof of concept, the same AF multimer run can be done with PBP4 and GspB to see if a more extended picture of the PBP4-GspB complex can be reached.

Following the reviewer’s suggestion, we have attempted modeling GpsB hexamer and the complex with FtsZ C-terminal domain. Despite extensive trials, AlphaFold could not model the GpsB hexamer (Author response image 2, left panel). Based on the crystal structures of the N-terminal domain dimer and C-terminal domain trimer of other GpsB homologs, an artistic rendering of the hexamer model is presented in Figure 5. For the interactions between GpsB and FtsZ C-terminal domain, AF was able to place the arginine-rich region in the canonical pocket of GpsB (Author response image 2 panel), although clashes were observed among the side chains at the binding interface reflecting the complexity and challenges of modeling. The best model was obtained when we focused on the last six residues of FtsZ (Figure 4—figure supplement 3), whose interactions with GpsB resemble those of PBP4 (shown in Figure 4B).

Reviewer #3 (Recommendations for the authors):The authors have adequately responded to the first round of reviews. I still believe that an FtsZ GTPase mutant would be the better control for the biochemical assay showing stimulation of GTPase activity by GpsB, but BSA is adequate.

We thank the reviewer for finding our revised manuscript satisfactory. At this moment, we do not have a FtsZ GTPase mutant. However, we plan to generate one to serve as a control in our follow-up studies.

I also still believe that a true examination of the functional importance of the hinge domain residues would require replacement of the native GpsB with these variants, followed by phenotypic characterization (though my understanding is that ∆gpsB phenotypes are subtile, but measurable nonetheless via e.g. morphology). The current reliance on overexpression toxicity (which was pointed out during the first round of reviews) makes interpretation of these data more difficult. That said, using a gpsB mutant for overexpression experiments is certainly an improvement over the first version of the manuscript.

Thank you, we also agree that using ∆*gpsB* mutant provided some clarity to our findings. We are in the process of submitting a manuscript revealing *B. subtilis* GpsB (that lacks the hinge region) also interacts with *B. subtilis* FtsZ. Thus, we think the hinge region is used for some specialized role in *S. aureus*.